# Mechanistic insights into CAM-induced disruption of HBV capsids revealed by all-atom MD simulations

Carolina Pérez-Segura[1], Boon Chong Goh[2†], Jodi A. Hadden-Perilla[1]*

**1** Department of Chemistry & Biochemistry, University of Delaware, Newark, Delaware, United States of America, **2** Antimicrobial Resistance Interdisciplinary Research Group, Singapore-Massachusetts Institute of Technology Alliance for Research and Technology Centre, Singapore

† Now at ArrowBiome, Singapore
* jhadden@udel.edu

## Abstract

Capsid assembly modulators (CAMs) represent a promising antiviral strategy against hepatitis B virus (HBV), but their effects on pre-formed capsids remain incompletely understood. Here, all-atom molecular dynamics (MD) simulations of intact HBV capsids complexed with prototypical CAM-As (HAP1, HAP18) and CAM-Es (AT130), reveal how structural changes induced by small molecule binding in the interdimer interfaces propagate through the shell lattice to yield global morphological consequences. Each quasi-equivalent interface exhibits a unique response: A sites, located within the pentameric capsomers, are unfilled in these systems and altered marginally by the presence of CAMs in neighboring interfaces. B sites are the most open and "CAM-ready," suggesting uptake requires minimal conformational perturbation on the local or global level. C sites emerge as hubs of allosteric control and the key drug target, as their occupancy creates local distortion that is broadcast to adjacent sites, driving capsid faceting and – in the case of CAM-As – the destabilization that precedes dissociation in favor of aberrant assembly. D sites, unfilled in these systems, act as structural sinks, absorbing distortions from adjacent interfaces within the hexameric capsomers. The extent of C site adjustment and the nature of D site counterbalance varies with CAM chemotype, highlighting the divergent effects of CAM-As versus CAM-Es. The tensegrity relationship between the four quasi-equivalent interfaces couples them into a global network for strain redistribution that is functionally allosteric, with CAM binding sites displaying signs of both positive and negative cooperativity. These new insights into HBV capsid dynamics clarify how CAMs alter them on the microsecond timescale and suggest that targeting strain redistribution in mature core particles could be leveraged therapeutically.

**Data availability statement:** The data supporting the conclusions of this work are available via Zenodo: https://doi.org/10.5281/zenodo.18098187.

**Funding:** This work was funded by the National Institutes of Health (NIH) through award P20GM104316-10 to J.A.H.-P and the University of Delaware. The funders had no role in study design, data collection and analysis, decision to publish, or preparation of the manuscript.

**Competing interests:** The authors have declared that no competing interests exist.

## Author summary

Hepatitis B is a major cause of chronic liver disease worldwide. The virus relies on a protein shell, called the capsid, to protect and deliver its genetic material to the host cell during infection. Some experimental drug molecules attack this shell, either forcing it to assemble incorrectly or breaking it apart after it has formed. To understand how these molecules work, we used powerful computer simulations to model the capsid at the level of individual atoms. We discovered that when molecules bind the capsid at certain sites, they create strain that spreads across the shell, sometimes leading to large distortions and instability. These insights explain how small molecules can disrupt the virus and point the way toward designing better antiviral therapies.

## Introduction

Hepatitis B virus (HBV) is an enveloped, partially double-stranded DNA virus from the *Hepadnaviridae* family. HBV infection remains a major global health concern, contributing to acute and chronic liver disease and increasing the risk of cirrhosis and hepatocellular carcinoma. According to the World Health Organization, more than 254 million people are chronically infected, with over one million deaths annually despite the availability of an effective vaccine [1]. While nucleos(t)ide analogs can suppress viral replication, they do not provide a cure, necessitating lifelong treatment for patients who cannot naturally clear the infection [2]. Alternative antiviral strategies include targeting the HBV capsid to inhibit its roles in particle assembly, pre-genomic RNA packaging, metabolic regulation of reverse transcription, and intracellular trafficking [3,4].

The capsid is an icosahedral protein shell that predominantly assembles with $T = 4$ symmetry from 240 copies of the core protein (Cp). Cp contains 183 residues (or 185 depending on genotype) partitioned into two domains: The N-terminal assembly domain (residues 1-149, Cp149), sufficient to drive Cp association into particles, and the arginine-rich C-terminal domain (CTD, residues 150-183), which binds the pre-genome and other essential host factors [4,5]. Although Cp149 lacks the CTD, it has been widely used in *in vitro* studies of capsid architecture and formation [6, 7]. Within the $T = 4$ lattice, Cp takes on four quasi-equivalent [8] conformations (A, B, C, D), arranged into 60 AB and 60 CD homodimers (Fig 1A). The Cp intradimer interface comprises a four-helix bundle [9]. The Cp interdimer interface arises from oligomerization of a base and capping chain during assembly [6,10]. These two interfaces are highly dynamic and allosterically coupled [11–13]. Five copies of A chains form pentameric capsomers, situated on the twelve icosahedral vertices. Two copies each of the B, C, and D chains form hexameric capsomers, situated on the 30 icosahedral edges [10]. Thus, pentamers represent fivefold symmetry axes and hexamers represent centers of twofold or quasi-sixfold symmetry. The trimeric junctures of hexamers situated on the 20 triangular faces of the icosahedron represent threefold symmetry axes. Owing to quasi-equivalence, there are four structurally distinct interdimer interfaces, which encompass hydrophobic pockets recognized by small

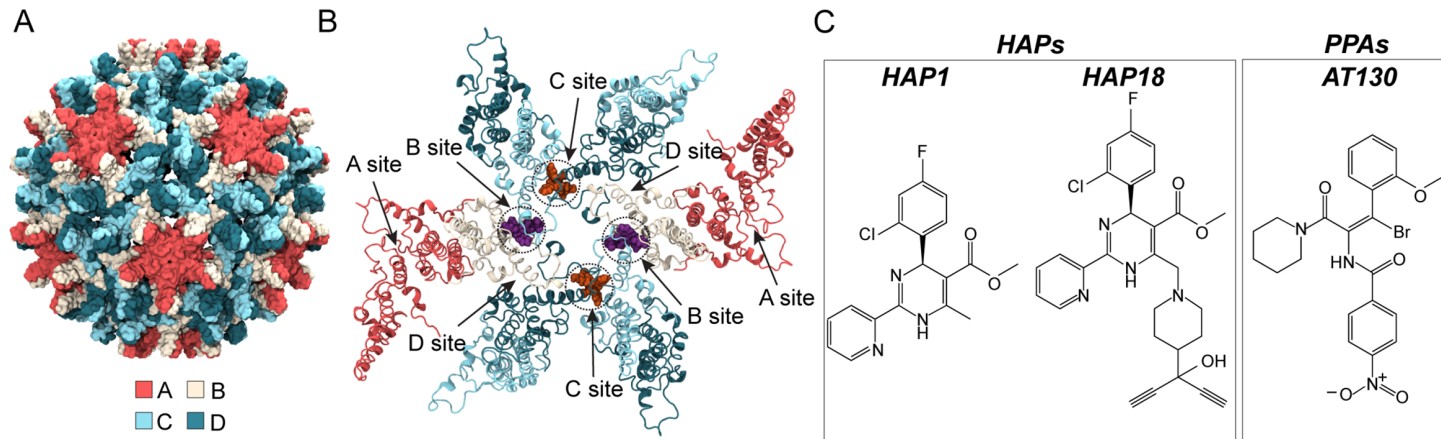

**Fig 1**. **HBV capsid structure and quasi-equivalent CAM binding sites.** (**A**) The $T = 4$ capsid incorporates 240 quasi-equivalent copies of Cp designated A (red), B (beige), C (cyan), and D (blue), arranged as 60 AB and 60 CD dimers. (**B**) Cp dimer interactions form four unique interdimer interfaces, each comprising a base chain capped by an adjacent chain. The CAM sites located within the interdimer interfaces are designated A (A-A interface), B (B-C interface), C (C-D interface), and D (D-B interface). The crystal structures of CAM-bound capsids examined here [14–16] exhibit occupancy in B and C sites, with example binding modes shown in purple and orange, respectively. (**C**) Chemical structures of CAMs under study: HAP1, HAP18, and AT130.

molecules (Fig 1B). These locations are designated: A site (A-A interface), B site (B-C interface), C site (C-D interface), D site (D-B interface).

Capsid assembly modulators (CAMs) are a diverse class of antivirals that interfere with HBV by targeting the interdimer interfaces [17]. By binding early-stage intermediates, CAMs alter the assembly process, leading to the production of particles that are non-infectious – either because they are structurally defective or devoid of genome. These molecules are classified based on their assembly outcomes: CAM-A molecules accelerate and misdirect assembly to generate *aberrant* polymerization products, including hexamer sheets or tubes [18,19]. CAM-E molecules accelerate capsid formation, sabotaging pre-genome packaging to generate morphologically normal but *empty* particles [20–22]. Known CAM-As include heteroaryldihydropyrimidines (HAPs), non-HAPs, and dibenzothiazepines (DBTs), while known CAM-Es include phenylpropenamides (PPAs), benzamides (BAs), sulfamoylbenzamides (SBAs), sulfamoylpyrrolamides (SPAs), and glyoxamoylpyrroloxamides (GLPs) [23–25]. Besides forming complexes with assembly intermediates, CAMs can also bind preformed capsids [14,15]. While uptake of CAM-Es does not compromise particle integrity, CAM-As can disrupt both empty and DNA-containing particles, causing them to dissociate [19]. Studies have shown that CAM-A binding induces local strain on the shell, which leads to global destabilization that can trigger disassembly of metastable capsids in favor of aberrant reassembly [14,26].

Here, all-atom molecular dynamics (MD) simulations are used to investigate the effects of prototypical CAM-As (HAP1, HAP18) and CAM-Es (the PPA AT130) on intact Cp149 capsids, in the absence of symmetry constraints. Selective removal of these CAMs (Fig 1C) from the B and C sites they are known to bind experimentally [14–16] probes the role of quasi-equivalence in the local conformational response of interdimer interfaces. Beyond simulations of capsid fragments and intermediates, this work reports on the integrated motions of the complete 120-dimer shell. By examining the intact capsid on the microsecond timescale, MD captures large-scale collective motions, redistribution of strain across the Cp lattice, and emergent properties that arise only within the fully assembled particle [27]. Owing to atomistic resolution, which is critical for studying capsid interactions with small molecules, [28] these simulations link chemical details of CAM recognition with quaternary changes that propagate to induce global morphological distortions. By combining local and

global structural perspectives, this work establishes how CAM occupancy influences pre-formed capsids and provides a framework for understanding divergent mechanisms of stabilization versus destabilization for different CAM chemotypes.

## Results

All-atom models of intact, CAM-bound HBV Cp149 capsids were prepared based on crystal structures of pre-formed capsids complexed with AT130 (PDB 4G93, 4.2 Å), [15] HAP1 (PDB 2G34, 5.0 Å), [14] and HAP18 (PDB 5D7Y, 3.9 Å) [16]. The AT130- and HAP18-bound capsid models contain CAMs in all B and C sites and are referred to as saturated. The HAP1-bound capsid model contains CAMs only in C sites, as the B site binding mode was not resolved and experimental coordinates are not available. Four models exploring variations on the HAP18-bound capsid were prepared by removing CAMs from all B sites (HAP18$_C$), all C sites (HAP18$_B$), B and C sites of one hemisphere (HAP18$_{hemi}$), and all sites (HAP18$_{apo}$). MD simulations of the HAP-affected capsids were performed on the microsecond timescale under physiological conditions, analogous to previously reported simulations of intact apo-form and AT130-bound capsids [29,30]. The simulation systems analyzed in this work are summarized in Table 1.

### Saturation of CAM sites increases capsid volume

Consistent with observations for the apo-form, [29] all CAM-affected capsids expanded as they relaxed from their crystallographic conformations (Fig 2A-C). Crystal structures indicate that HBV capsids are slightly larger with CAMs complexed, [14–16] and MD simulations reveal that the increase in shell volume is conferred by simultaneous occupancy of B and C sites. The AT130- and HAP18-bound capsids, both saturated with CAMs, converged to a similar volume ~3% larger than the apo-form (Fig 2A). The HAP1-bound capsid, which was simulated with CAMs only in C sites, converged to the same volume as the apo-form (Fig 2A). These two size states are differentiated by the presence of CAMs in B sites. However, capsid volume is not necessarily an indicator of capsid shape or local conformational adjustments that accommodate CAMs.

While the apo-form and AT130-bound capsids reached their equilibrium shell volumes within 0.1 $\mu$s, the capsids complexed with HAPs required hundreds of nanoseconds to relax to physiological conditions (Fig 2A). B site occupancy was observed in the HAP1-bound capsid crystal, [14] although CAM coordinates could not be resolved there. As such, the simulated HAP1-bound capsid initially trended toward the size state exemplified by the saturated AT130- and HAP18-bound capsids, consistent with their crystallographic conformations exhibiting similar shell volumes. However, structural adaptation of the HAP1-bound model to vacant B sites during simulation required almost 0.4 $\mu$s (Fig 2A), reflecting the timescale necessary for local adjustments across 60 interdependent interdimer interfaces to converge globally.

Remarkably, the simulated HAP18-bound capsid showed initial signs of reaching a third size state, ~5% larger than the apo-form, before trending toward equilibrium (Fig 2A). This behavior suggests other environmental factors at play

**Table 1**. Summary of intact HBV capsid MD simulations.

| System | PDB | Occupied sites† | CAMs/capsid‡ |
|---|---|---|---|
| apo-form | 2G33 | - | - |
| AT130 | 4G93 | B and C | 120 |
| HAP1 | 2G34 | C | 60 |
| HAP18 | 5D7Y | B and C | 120 |
| HAP18$_B$ | 5D7Y | B | 60 |
| HAP18$_C$ | 5D7Y | C | 60 |
| HAP18$_{hemi}$ | 5D7Y | B and C, hemisphere | 60 |
| HAP18$_{apo}$ | 5D7Y | - | - |

†Remaining CAMs removed in HAP18 variations.
‡120 CAMs (all B/C sites filled) is referred to as saturated.

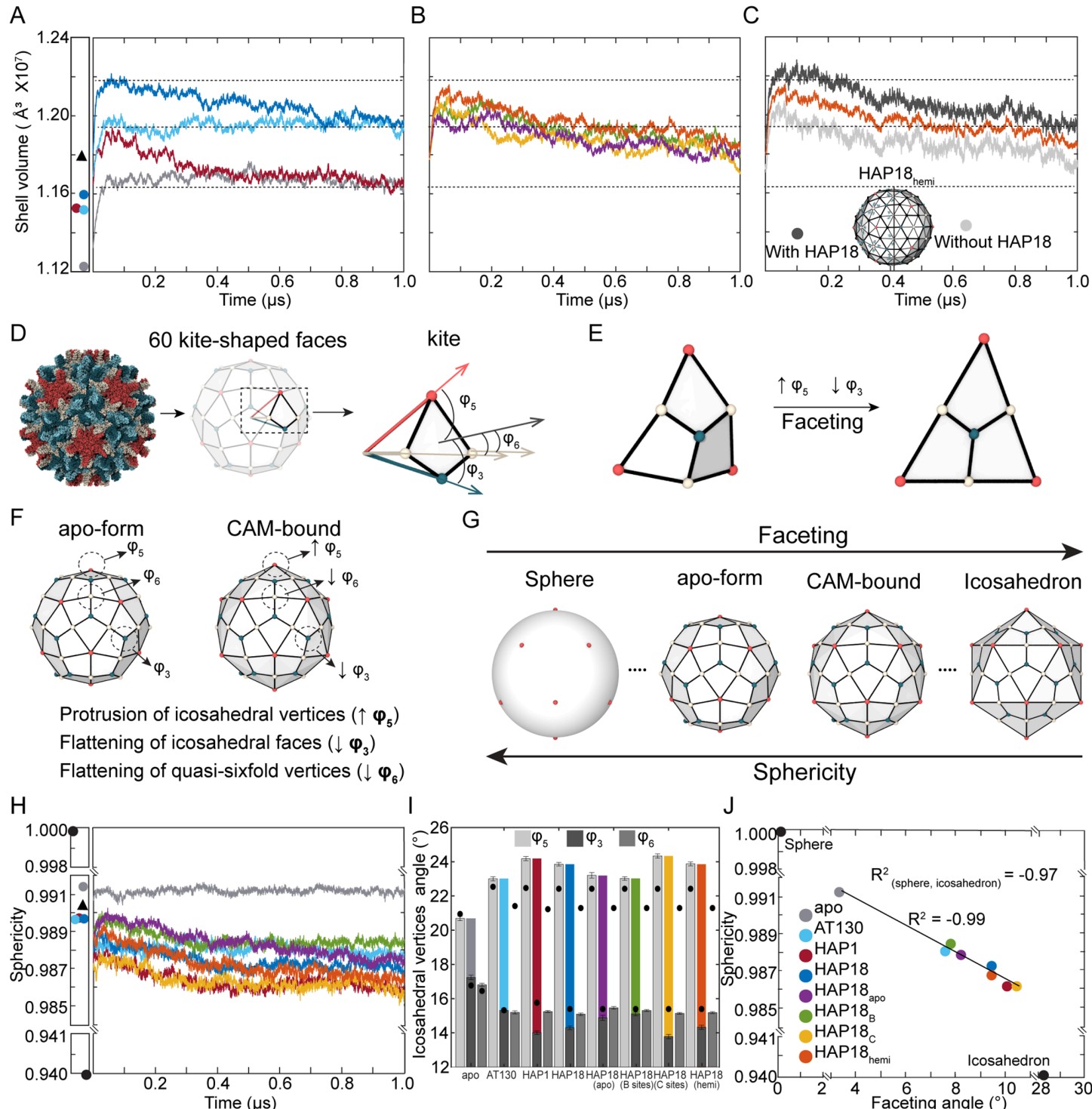

**Fig 2. Morphological properties of simulated HBV capsids.** Evolution of capsid shell volume over time for (**A**) apo-form, AT130-bound, HAP1-bound, HAP18-bound systems, (**B**) HAP18$_{apo}$, HAP18$_B$, HAP18$_C$, HAP18$_{hemi}$ systems. Shell volume of corresponding crystal structures indicated by scatter plot (left). (**C**) Capsid shell volume of HAP18$_{hemi}$ decomposed by hemisphere, where values of each half-shell are normalized for comparison to intact systems. (**D**) Angles characterizing average orientation of asymmetric units (generalized as kites) with respect to nearby fivefold ($\varphi_5$), threefold ($\varphi_3$), and quasi-sixfold ($\varphi_6$) vertices. (**E**) The relative protrusion of fivefolds versus flattening of threefolds results in faceting, (**F**) which occurs in the presence of bound CAMs and (**G**) alters capsid morphology. (**H**) Evolution of capsid sphericity over time. Sphericity values for corresponding crystal structures,

as well as for perfect sphere and regular icosahedron, indicated by scatter plot (left). (**I**) Average values of $\varphi_5$, $\varphi_3$, $\varphi_5$, with faceting angle $\varphi_5 - \varphi_3$ shown in color. (**J**) Sphericity and faceting angle display a linear correlation for the eight systems under study, indicating that CAM-bound capsids are less spherical because they are more faceted in morphology rather than simply asymmetric. Triangular points in scatter plots (panels **A,H**) represent data for a DBT1-bound cryo-EM structure [26]. Average values used in panels **I,J** are based on the last 500 ns of simulation. Colors used in plots are defined in panel **J** legend.

in the HAP18-bound crystal that were not reproduced during simulation, which required over 0.7 $\mu$s to relax out. Variations on the HAP18 system with CAMs removed also exhibited this more gradual trend of relaxation (Fig 2B). With further sampling, the HAP18$_{apo}$ capsid should equilibrate to a structural ensemble indistinguishable from that of the apo-form. Likewise, both HAP18$_{apo}$ and HAP18$_C$ capsids would be expected to settle on the size state exhibited by the apo-form and HAP1-bound capsids, given that they lack B site occupancy. However, the relaxation trends of the HAP18$_B$ and HAP18$_{hemi}$ capsids suggest that volume expansion is induced by saturation of both B and C sites. Given that HAP18 is a large and highly distortive CAM, the finding that simultaneous B/C site occupancy is responsible for capsid size increase, rather than B or C sites alone, likely holds for smaller CAMs, as well as less structurally disruptive chemotypes.

Although the HAP18$_B$, HAP18$_C$, and HAP18$_{hemi}$ capsids – which each contain 60 CAM copies – show roughly similar relaxation trends (Fig 2B), further inspection revealed that shell volume calculated for the HAP18$_{hemi}$ capsid represents contributions from two disparate hemispheres. The hemisphere with CAMs complexed exhibits a larger volume, consistent with that of the saturated HAP18-bound capsid, while the hemisphere without CAMs exhibits a smaller volume more akin to that of the HAP18$_{apo}$ capsid (Fig 2C). Thus, structural adjustments required to accommodate CAMs in B and C sites do not propagate across the entire shell, although they do give rise to meaningful local effects described below. Instead, saturation of one hemisphere with CAMs induces a subtle ovoid distortion of the global capsid morphology, and overall capsid volume is a product of the collective response of the interdimer interfaces to bound CAMs.

### CAMs, especially in C sites, induce capsid faceting

HAP molecules are known, both experimentally [14,31] and computationally [28] to induce faceting in HBV capsid structures. Faceting is determined by the orientation of the asymmetric unit, representing one third of an icosahedral face and generalized here as a kite-shaped polygon (Fig 2D). Tilting of the asymmetric unit relative to nearby symmetry axes controls the appearance of triangular facets (Fig 2E) and can be quantified by measuring the resulting protrusion of icosahedral vertices ($\varphi_5$) versus flattening of icosahedral faces ($\varphi_3$) [28] (Fig 2F). The $\varphi_5 - \varphi_3$ disparity, referred to as the faceting angle, indicates the extent to which the capsid's morphology approaches that of a regular icosahedron (Fig 2G). While the apo-form capsid was observed to remain highly spherical during simulation, showing little change in this property compared to the crystal structure, [29] all CAM-affected capsids exhibit reduced sphericity, deviating more dramatically from their crystallographic conformations (Fig 2H). The CAM-affected capsids are less spherical because they are more faceted. Besides increased $\varphi_5$ and decreased $\varphi_3$, leading to larger faceting angles, the CAM-affected capsids exhibit reduced protrusion at quasi-sixfold vertices (decreased $\varphi_6$), consistent with structural changes that favor hexamers and their trimeric junctures at the expense of pentamers (Fig 2I).

The sphericity scale ranges from [0-1] and remains above 0.9 for many geometric shapes that are not obviously sphere-like. Sphericity is also independent of symmetry. Thus, the metric does not at first appear sensitive to subtle structural adjustments arising from the binding of small molecules to large macromolecular assemblies, even if those perturbations are numerous and additive. However, while the differences in mean sphericity between simulated capsids are small, the corresponding frequency distributions show limited overlap, particularly between the apo-form and CAM-bound systems (S1A Fig). Based on the Kruskal–Wallis [32] and Dunn–Bonferroni [33–35] tests, the differences are statistically significant for all but two systems: HAP1 and HAP18$_C$, which contain CAM-As exclusively in C sites (S1B Fig).

These results indicate that the CAM-induced morphological changes captured by MD are meaningful with respect to the apo-form capsid, as well as reproducible across independent simulations of capsid-HAP systems, supporting the validity of findings in the absence of data collection replicates.

Sphericity can be directly correlated with the extent of faceting, where the linear regression including analogous points for a perfect sphere and regular icosahedron has a very high coefficient of determination ($R^2 = 0.97$, Fig 2J). Excluding these geometric extremes, the correlation between sphericity and faceting for the simulated capsids is yet stronger ($R^2 = 0.99$). Interestingly, the CAM-affected capsids cluster into three groups along the regression trendline, revealing the response of capsid shape to CAM occupancy. The HAP1 and HAP18$_C$ systems, which contain CAMs only in C sites, exhibit the most pronounced faceting. The HAP18 and HAP18$_{hemi}$ systems, which contain CAMs in both B and C sites in at least half of the capsid, show slightly less faceting. Less faceted still are the AT130, HAP18$_B$, and HAP18$_{apo}$ systems. Together, these observations indicate that the binding of HAPs in C sites is most responsible for inducing morphological disruptions. The additional binding of HAPs in B sites attenuates this effect. Given the minimal faceting observed for the AT130 and HAP18$_B$ systems, and that assembly in the presence of AT130 is known to produce capsids rather than aberrant structures, [15,22] it may be that uptake of CAM-As exclusively in B sites would not lead to dissociation of pre-formed shells. The finding that capsids saturated with CAM-Es exhibit enhanced volume, reduced sphericity, and minor morphological distortions relative to the apo-form is consistent with predictions from coarse-grain studies of HBV Cp assembly [36].

## CAM-As promote hexamer bending and asymmetry

Capsid faceting arises from local structural adjustments to accommodate CAMs within the interdimer interfaces of hexamers. In the $T = 4$ capsid, hexamers are centered on the icosahedral edges, which connect adjacent pentamers on the icosahedral vertices. Besides flattening the trimeric junctures between hexamers, CAM-As cause hexamers to bend or crease along the icosahedral edges, emphasizing the triangular faces of the icosahedron as facets. Although cryo-EM studies of HAP-bound capsids reported flattening of hexamers concomitant with faceting, [31] this particular response occurs exclusively along the the AB-BA dimer axes (the icosahedral edges), but not along the DC-CD and CD-DC axes. Curvature along all three dimensions of the quasi-sixfold vertices can be characterized by the relative orientation of these paired dimers across the centers of hexamers ($\phi_{BA}$ for AB-BA, $\phi_{CD}$ for DC-CD, and $\phi_{DC}$ for CD-DC, Fig 3A).

Fig 3B illustrates the two extremes of hexamer conformation, as observed during MD simulations, and the structural consequences for the triangular faces they join. The subtle curve of the apo-form capsid shell versus the faceted shape of the HAP1-bound capsid shell is primarily distinguished by decreased $\phi_{BA}$ and increased $\phi_{DC}$, owing to CAM occupancy in C sites. The former angular shift reflects true flattening along the AB-BA axes and accounts for reduced protrusion at the quasi-sixfolds, concomitant with enhanced protrusion at the fivefolds. The latter reflects tilting of dimers to confer acute curvature rather than planarity along the CD-DC axes, concomitant with flattening at the threefolds. This bending or creasing of hexamers, which is most apparent when $\phi_{CD}$ also samples larger values (Fig 3B, left), sharpens the icosahedral twofolds and accounts for the pronounced icosahedral edges observed in micrographs of HAP-bound capsids [31]. The altered orientation of CD-DC dimer pairs leads to distortion of hexameric pores (Fig 3B, right), which can expand asymmetrically even in apo-form capsids [37]. Changes in pore topology had no discernible impact on solvent exchange rates (S2 Fig).

Comparing distributions of $\phi_{BA}$, $\phi_{CD}$, and $\phi_{DC}$, MD simulations demonstrate that hexamers explore a range of conformations, as well as CAM-induced shifts, beyond those captured in crystal structures. For apo-form hexamers, the crystallographic state is roughly representative of observations by MD (Fig 3C). The same is true for AT130-bound hexamers, although more flattened AB-BA dimer pair configurations are preferred (Fig 3D). Notably, all CAM-affected capsids explore smaller $\phi_{BA}$ values than are possible in the apo-form, which accounts for their reduced sphericity. Significant overlap between $\phi_{CD}$ and $\phi_{DC}$ distributions in both the apo-form and AT130-bound systems indicates ensemble-level

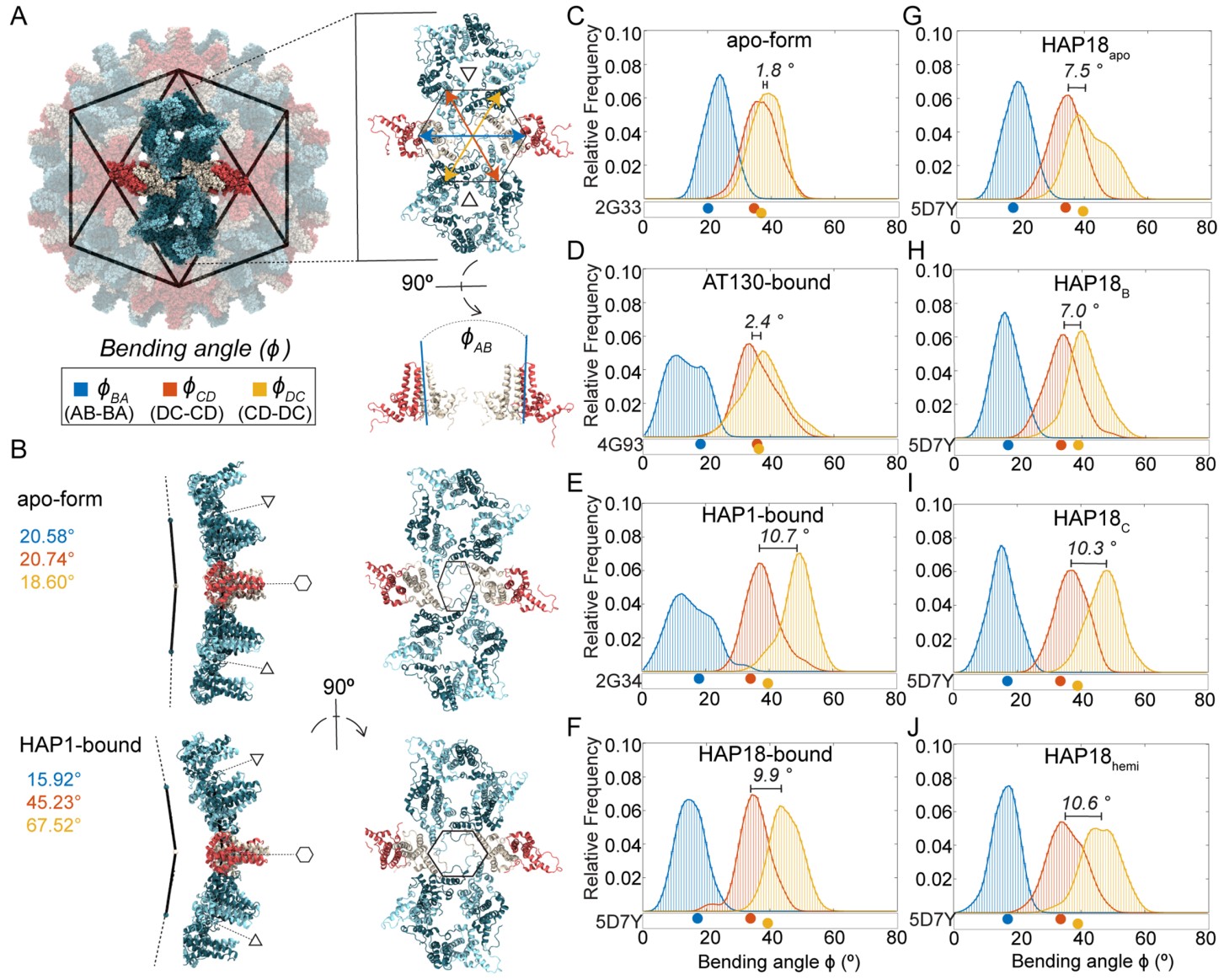

**Fig 3. Hexamer response to bound CAMs in simulated HBV capsids.** (**A**) AB dimers lie along icosahedral edges and CD dimers are grouped on the triangular faces. The relative orientation of paired dimers across the hexamer ($\phi_{BA}$, $\phi_{CD}$, $\phi_{DC}$) capture the structural origins of its curvature. Angle values are shown in blue, orange, and yellow, respectively. (**B**) Hexamer conformations extracted from MD simulations. Comparing the faceted HAP1-bound capsid to the apo-form: Smaller values of $\phi_{BA}$ indicate hexamer flattening along the AB-BA axis, concomitant with protrusion at the fivefold vertices. Larger values of $\phi_{DC}$ indicate hexamer bending along the CD-DC axis, concomitant with flattening at the threefold vertices. The result is more acute curvature of the hexamer, a sharpened crease along the icosahedral edge, and concomitant distortion of the quasi-sixfold pore. The HAP1-bound conformation shown represents an extreme example, selected for visual clarity, and its like is not found within the apo-form capsid ensemble. Hexamer bending angle distributions for (**C-F**) apo-form, AT130-bound, HAP1-bound, HAP18-bound systems, and (**G-J**) HAP18$_{apo}$, HAP18$_B$, HAP18$_C$, HAP18$_{hemi}$ systems. Distributions are based on the last 500 ns of simulation. The shift between $\phi_{CD}$ and $\phi_{DC}$ ensembles is marked on each plot. Points below distributions indicate angles measured for corresponding crystal structures.

symmetry between DC-CD and CD-DC dimer pairs, without implying synchronized bending within or between these pairs.

While HAP-bound capsid crystals would suggest such symmetry persists regardless of CAMs, MD simulations reveal otherwise. Both the HAP1 and HAP18 systems deviate from their crystallographic state following a shift to larger $\phi_{DC}$

values, leading to pronounced separation between $\phi_{CD}$ and $\phi_{DC}$ distributions (Fig 3E-F). This response is not explained by structural adaptation of HAP1-bound hexamers to vacant B sites, since HAP18-saturated hexamers exhibit analogous behavior upon relaxation. Interestingly, AB-BA dimer pair dynamics in the HAP1-bound case are more similar to the AT130-bound than any other system, despite the fact that these represent the most and least faceted CAM-bound capsids, respectively.

Simulations with varying HAP18 occupancies indicate that increased $\phi_{DC}$ arises primarily from the presence of CAM-As in C sites, consistent with the role of CD-DC dimer pair orientation in capsid faceting: HAP18$_{apo}$ samples large $\phi_{DC}$ values owing to its starting conformation, but shows evidence of trending toward the apo-form distribution over time (Fig 3G). HAP18$_B$ and HAP18$_C$ likewise sample large $\phi_{DC}$ values (Fig 3H-I), but the hexamer asymmetry implied by their $\phi_{CD}$ versus $\phi_{DC}$ ensemble disparity is greater in the case of C site occupancy. Remarkably, the distribution shift for HAP18$_{hemi}$ (Fig 3J) is similar in magnitude (~10°) to all other systems containing HAPs in C sites, despite having small molecules complexed in only half of the shell. On the other hand, HAP18$_{apo}$ and HAP18$_B$ exhibit a smaller distribution shift (~7°), more consistent with the crystal structure, but apparently the result of empty C sites. That the crystallographic state falls within the range of equilibrium dynamics reported by MD, yet fails to capture the extent of hexamer distortion induced by bound HAPs may relate to its use of the Cp150 construct, [38] which confers shell stability by cross-linking the C-termini of Cp149 dimers. Experimental structural studies examining the effects of CAM-As of pre-formed capsids commonly employ Cp150 [14,16].

## CAMs promote planarity along hexamer-hexamer junctures

The influence of CAMs on hexamer conformation propagates to the trimeric junctures between hexamers and the triangular pores thus defined (Fig 4). Owing to the $T = 4$ capsid architecture, the B, C, and D quasi-equivalent CAM sites lie nearly along the icosahedral edges. Hexamers that become bent or creased along these edges to accommodate CAMs necessitate flatter junctures to preserve inter-capsomer contact on the icosahedral faces and maintain a closed shell. That is, increased protrusion at fivefolds and bendeing along twofolds in faceted capsids is – to some extent – compensated by a reduction of curvature at threefolds. The molecular details underlying curvature versus planarity of hexamer junctures in intact $T = 4$ capsids arise from the behavior of CD dimers. Notably, their interdimer interfaces encompass the C sites recognized by CAMs, which appear to have the most dramatic impact on pre-formed capsid morphology.

Fig 4 illustrates the two extremes of triangular pore conformation as observed during MD simulations. The relative orientation of CD dimers across the C site can be quantified by the spike and base angles ($\theta_{spike}$, $\theta_{base}$) [39] between them. Flattening of the trimeric juncture in response to bound HAPs is reflected by decreased $\theta_{spike(C)}$ relative to the apo-form (Fig 5A). Average $\theta_{base(C)}$ fluctuates around ~60° regardless of HAPs, with triangular vertices summing to the required 180°. Besides enhancing planarity of the icosahedral face, tilting of CD dimers toward the threefold vertex alters the topology of the enclosed pore, although without a discernible impact on solvent exchange rates (S2 Fig). In the most extreme cases of $\theta_{spike(C)}$ adjustment, CD spikes can become essentially parallel (Fig 4A), though on average HAPs induce a ~20° orientational change with respect to the preferred apo-form conformation (Fig 4B). Importantly, owing to the capsid's ability to accommodate local strain by distributing it across neighboring subunits, flattening does not necessarily occur simultaneously across all three interdimer interfaces of a given trimeric juncture. Further, although uncommon, curved junctures were observed in HAP-bound capsids and flat junctures were observed in the apo-form during MD simulations. As local phenomena, conformations that are rare for the respective morphological states can be tolerated because the capsid is highly dynamic and instantaneously asymmetric [29]. In general, global symmetry of the icosahedral shell is an ensemble property.

## CAMs promote curvature of pentamer-hexamer junctures

Besides triangular pores formed by the juncture of three hexamers, the $T = 4$ capsid also contains so-called pseudo-triangular pores formed by the juncture of two hexamers and a pentamer (Fig 6A). The pseudo-triangular pores are

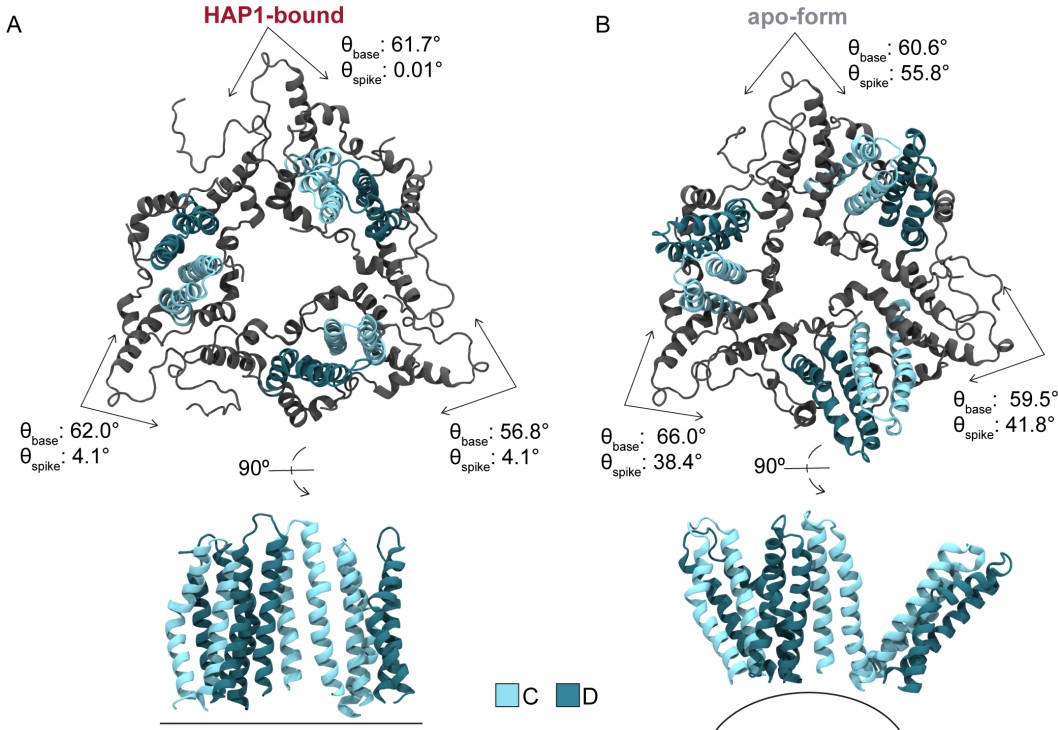

**Fig 4**. **Trimeric juncture response to bound CAMs in simulated HBV capsids.** Trimer-of-dimer (CD-CD-CD) conformations extracted from MD simulations illustrate the morphological impact of spike angle ($\theta_{spike(C)}$) changes induced by CAM binding. These interdimer interfaces correspond to quasi-equivalent C sites, whose dynamics are most responsible for conformational variability of the hexamer-hexamer interface. Comparing the faceted HAP1-bound capsid (**A**) to the apo-form (**B**): The former exhibits smaller $\theta_{spike(C)}$, corresponding to a flattening of the trimeric juncture between hexamers on the icosahedral faces. Dimers tilt inward toward the triangular pore, altering pore topology, and (in the most extreme cases) causing the relative orientation of CD spikes to become parallel. In contrast, the apo-form exhibits larger $\theta_{spike(C)}$, corresponding to curvature along the icosahedral face rarely observed in HAP-bound systems.

located in the centers of capsid asymmetric units, such that three copies are arranged on each triangular face of the icosahedron. Thus, while there are 20 triangular junctures, there are 60 pseudo-triangular junctures. Whereas the conformational behavior of the former is controlled by the interdimer interfaces that encompass C sites, the latter involves the remaining three quasi-equivalent sites: A, B, and D. Because each pseudo-trimeric juncture includes two icosahedral edges and one third of a face, the response of the subunit to bound CAMs is a consequence of edge creasing and face flattening. As the asymmetric unit reorients during capsid faceting, the pseudo-trimeric juncture controls the propagation of structural perturbations originating in the hexamers to the pentamers.

The relative orientation of dimers across the CAM sites comprising the pseudo-trimeric juncture can likewise be quantified by the spike and base angles ($\theta_{spike}$, $\theta_{base}$) [39] between them. Remarkably, HAP occupancy in B sites has minimal impact on the wide range of $\theta_{spike(B)}$ explored but increases the preferred value by ~5°, consistent with trends apparent in the apo-form and HAP18-bound crystals [14,16] (Fig 5B). The preferred $\theta_{base(B)}$ remains roughly constant. Considering the relatively small change in B sites when complexed with HAPs, the flattening of AB-BA dimer pairs across hexamers (reduced average $\phi_{BA}$, Fig 3) stems from the comparatively dramatic shifts within occupied C sites. That is, the significant distortion of filled C sites has limited impact on adjacent B sites despite their sharing of a C chain. Instead, distortion in C sites is accommodated primarily via referral to adjacent D sites, given their sharing of a D chain. This observation further accounts for the consistent behavior of DC-CD dimer pairs across hexamers (invariant average $\phi_{CD}$, Fig 3) regardless of

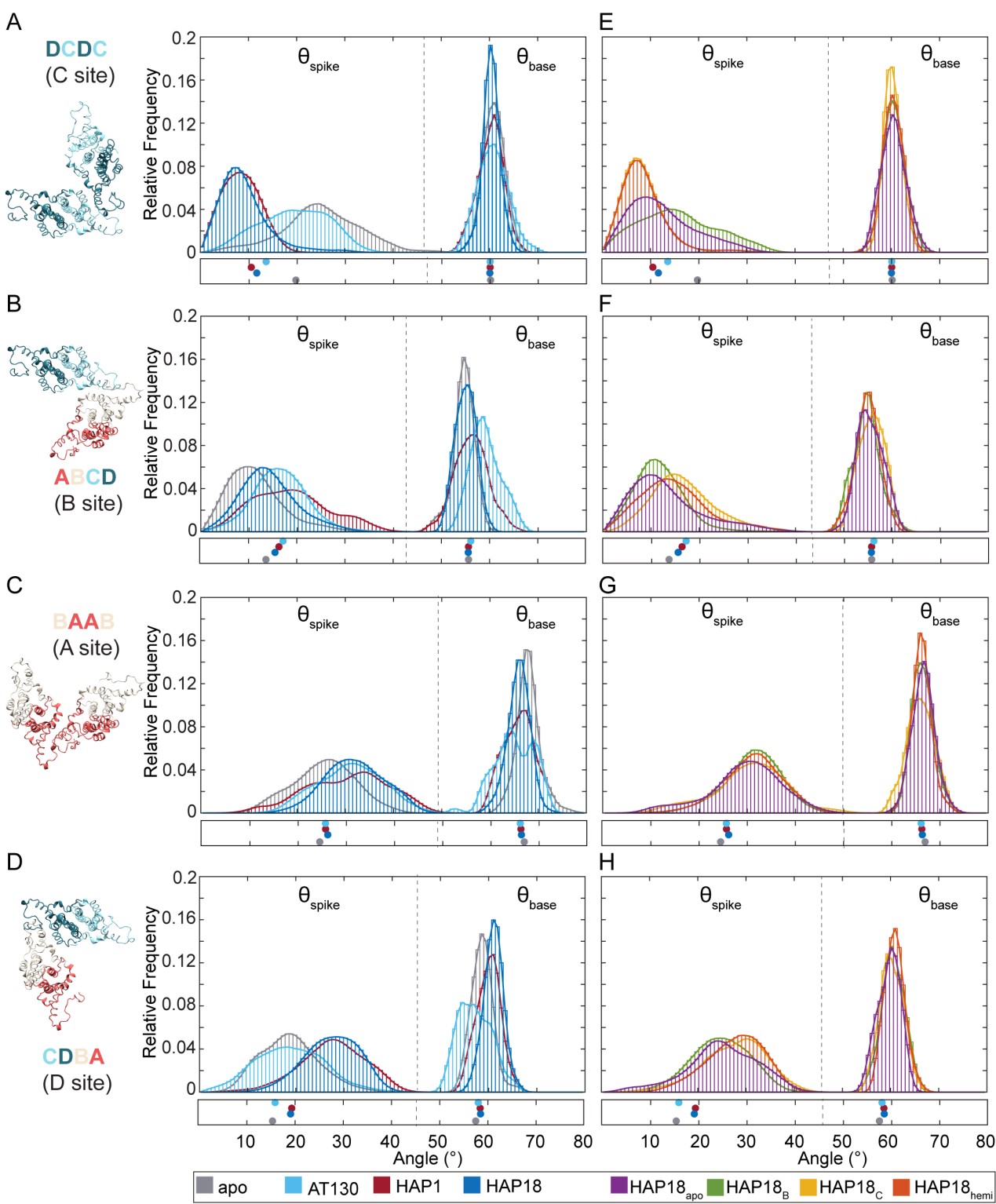

**Fig 5. Interdimer interface response to bound CAMs in simulated HBV capsids.** Relative orientations of adjacent dimers, which define CAM binding sites, are characterized by spike ($\theta_{spike}$) and base ($\theta_{base}$) angles. Distributions for quasi-equivalent C site (DC-DC dimers), B site (AB-CD dimers), D site (CD-BA dimers), A site (BA-AB dimers) shown for (**A-D**) apo-form, AT130-bound, HAP1-bound, HAP18-bound systems, and (**E-H**) HAP18$_{apo}$, HAP18$_{B}$, HAP18$_{C}$, HAP18$_{hemi}$ systems. Dashed lines partition $\theta_{spike}$ versus $\theta_{base}$ data. Distributions are based on the last 500 ns of simulation. Points below indicate values measured for corresponding crystal structures.

CAM occupancy. That is, adjustments in C sites propagate to D sites (via motion of D chains) to minimally affect B sites (via motion of B chains), thus flattening AB-BA pairs without altering the relative orientation of DC-CD pairs.

Although A and D sites are vacant in pre-formed HAP-bound capsids, their response to HAP complexation in the other two quasi-equivalent interfaces is analogous to that of occupied B sites. Intermediate to large orientations of CD dimer spikes become essentially disallowed in C sites containing HAPs, dramatically altering the shape of the trimeric junctures (Fig 5A). Nevertheless, A, B, and D sites retain the majority of their range of motion and adaptability, with the preferred values of $\theta_{spike(A)}$ and $\theta_{spike(D)}$ increasing by ~5° and ~10°, respectively (Fig 5C-D). Together, the positive adjustments (curvature) of A, B, and D sites in the pseudo-trimeric junctures (collective shift of 5° + 5° + 10° ≈ [+]20°, Fig 5B-D, left) compensate for the negative adjustments (flattening) of C sites in the trimeric junctures (collective shift of ≈ [−]20°, Fig 5A, left). While average $\theta_{base(B)}$ is constant regardless of HAP occupancy, $\theta_{base(D)}$ and $\theta_{base(A)}$ respond asymmetrically, exhibiting a minimal increase versus decrease when saturated with HAPs. These subtle trends are apparent in the HAP18 crystal structure, [16] with pseudo-triangular vertices still summing to the required 180° (Fig 5B-D).

Fig 6B-D illustrates the range of $\theta_{spike(D)}$ conformations, as observed during MD simulations, which have the most significant influence on the pseudo-trimer geometry. Small $\theta_{spike(D)}$ values correspond to a planar state of the pentamer-hexamer interface, in which AB and CD dimer spikes are roughly parallel, similar to the flattened hexamer-hexamer junctures that can be observed when HAPs occupy C sites (Fig 6B). Intermediate $\theta_{spike(D)}$ values are preferred and correspond to subtle curvature along the juncture (Fig 6C). Curvature increases with bound HAPs as distortion in filled C sites propagates to D sites. Large $\theta_{spike(D)}$ values correspond to highly curved states of the pentamer-hexamer interface, which are relatively rare in the apo-form capsid, but compensate for planar states of the hexamer-hexamer junctures in the event of faceting (Fig 6D). These D site adjustments in the presence of HAPs underlie the bending of CD-DC dimer pairs across hexamers (increased average $\phi_{DC}$, Fig 3). That is, flexibility across D sites accommodates strain induced by filled C sites,

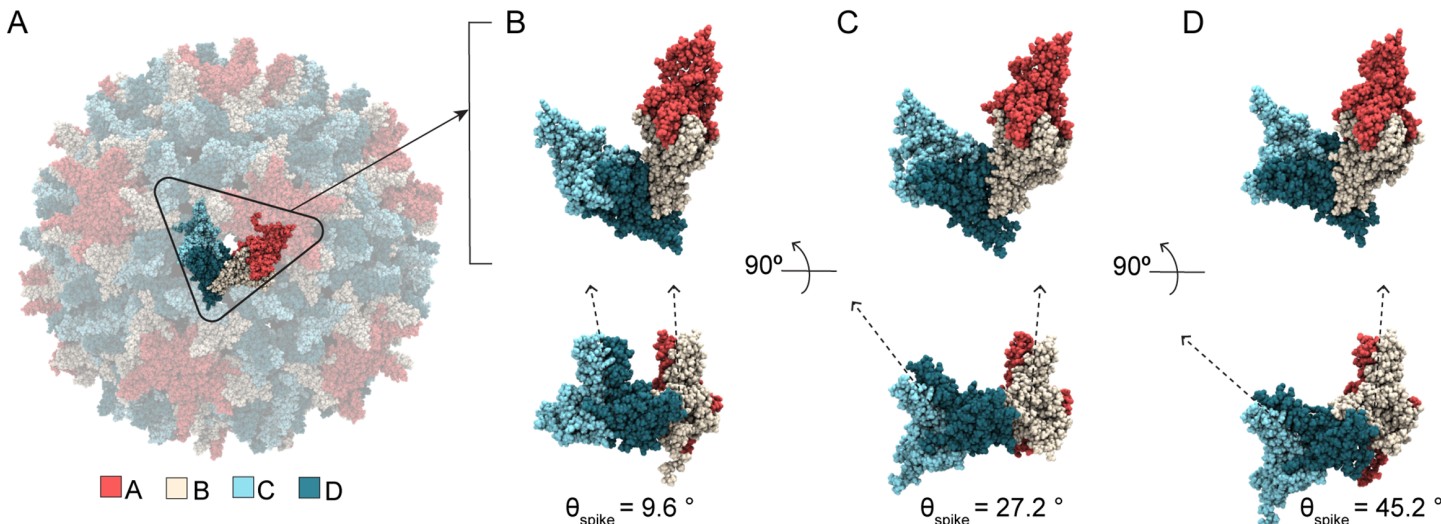

**Fig 6**. **Pseudo-trimeric juncture response to bound CAMs in simulated HBV capsids.** (A) Asymmetric unit (CD-BA) conformations extracted from MD simulations illustrate the morphological impact of spike angle ($\theta_{spike(D)}$) changes induced by CAM binding. This definition of the asymmetric unit corresponds to the quasi-equivalent D site, whose dynamics are most responsible for conformational variability of the pentamer-hexamer interface. (B) Small $\theta_{spike(D)}$ aligns AB-DC in the same plane, giving rise to a flat conformation where dimer spikes are nearly parallel. (C) Intermediate $\theta_{spike(D)}$ leads to subtly curved conformations, which are most common. Representative structure for HAP1-bound ensemble peak (Fig 5). (D) Large $\theta_{spike(D)}$ leads to dramatically curved conformations, related to the bent or creased hexamers observed in faceted capsids. All structures show changes in CD dimer orientation relative to AB dimer.

which ultimately maintains the closed shell via shifts in the icosahedral geometry. These changes alter the topology of the pseudo-triangular pore, without discernible impact on solvent exchange rates (S2 Fig).

**CAM effects on free versus capsid-incorporated subunits**

Fig 5 shows MD-derived distributions of spike and base angles ($\theta_{\text{spike}}$, $\theta_{\text{base}}$) [39] for all quasi-equivalent interfaces of the $T = 4$ capsid. Distribution overlap scores, ranging from 0→1 for maximally different to identical [40] are given in S3 Fig. As describe above, large versus small $\theta_{\text{spike}}$ reflect more curved versus flat geometries, respectively. MD simulations of isolated dimers-of-dimers – the earliest capsid assembly intermediates and the minimal CAM sites – show that they tend to adopt curved conformations as apo-form or with CAM-Es bound (*e.g.,* AT130), versus flat conformations with CAM-As bound (*e.g.,* HAPs) [39,41]. This observation has led to the hypothesis that curved conformations are "assembly active" and nucleate productive assembly pathways [7,39,41]. MD simulations of intact capsids extend these results to examine quasi-equivalence and constraints of the icosahedral shell as factors influencing interdimer orientation and response to CAMs. As implied by crystal structures, the conformational space sampled by capsid-incorporated dimers-of-dimers is limited compared to the analogous intermediates. Plots of standard deviation ellipses (SDEs) are presented in S4 Fig to facilitate comparisons with $\theta_{\text{spike}}$, $\theta_{\text{base}}$ space previously determined for intermediates using all-atom MD simulations [39].

While unassembled dimers-of-dimers show significant variability in $\theta_{\text{base}}$, unassembled trimers-of-dimers become locked into values ~60° appropriate for triangles [39]. As icosahedra composed of triangular and pseudo-triangular subunits, this behavior is inherited by intact capsids, although the distribution widths vary with local environment, CAM occupancy, and CAM class. Interestingly, the AT130-bound capsid explored the widest range of $\theta_{\text{base}}$ states, even exhibiting non-Gaussian behavior and reduced overlap scores compared to other systems for A, B, and D sites that define the pseudo-triangular pores (Figs 5A-D and S3). This outcome may be a result of the relative adaptability of the AT130 binding mode (S5 Fig), and thus its filled interdimer interfaces, which even upon saturation allow formation of the closed shell. Among HAP-affected capsids, $\theta_{\text{base}}$ is highly similar (overlap >0.7, Figs 5A-H and S3) indicating conserved dynamics, particularly for C sites where the symmetry of triangular pores enforce consensus.

Within the intact capsid, $\theta_{\text{spike}}$ is much more responsive to the presence of CAMs, and MD-derived distributions reflect trends apparent in crystal structures [14–16]. While HAP occupancy flattens C sites, AT130 occupancy allows curvature more akin to the apo-form (Fig 5A), consistent with behavior observed in free dimers-of-dimers [39]. Interestingly, HAP18$_{\text{apo}}$ and HAP18$_{\text{B}}$ – which lack CAMs in C sites – retain probability for curvature across this interdimer interface (Fig 5E), explaining why the faceting behavior of these systems in on par with the AT130-bound capsid. Systems that contain HAPs in C sites, including the half-saturated HAP18$_{\text{hemi}}$, exhibit nearly identical ensembles in terms of CD dimer orientations (overlap >0.9, Figs 5A,E and S3). Binding of a single HAP is sufficient to induce flattening in free trimers-of-dimers, [39] but within the constraints of the capsid lattice, it appears that partial saturation only sustains this preference if the subunit is incorporated at a trimeric juncture rather than pseudo-trimeric juncture. Importantly, flat conformations – although they may be less assembly-competent than curved conformations – are not incompatible with a closed shell, as long as they are compensated for elsewhere in the lattice. Indeed, apo-form B and D sites are significantly flatter within intact capsids (Fig 5B,D) than within free intermediates whose interdimer angles are thought to render them assembly-active, [39,41] suggesting that capsid-incorporated dimers are under tension.

Unlike C sites, quasi-equivalent B sites exhibit minimal change in $\theta_{\text{spike}}$ in response to CAM occupancy. As the most naturally flat interdimer interfaces in the absence of CAMs – a consequence of their position within the icosahedral lattice, B sites may be primed for small molecule binding and require the fewest conformational adjustments to receive them. This possibility aligns with observations from pre-formed capsid crystal structures, which indicate that B chains undergo minimal rearrangements upon HAP complexation [16]. Counterintuitively, CAMs induce curvature in B sites; however, consistent with unassembled dimers-of-dimers, [39] AT130-bound interfaces are more curved than HAP-bound interfaces

(Fig 5B). Distribution overlap remains >0.5 for all simulated capsids, reflecting relatively few differences in B site behavior (S3 Fig). The HAP18$_{apo}$ and HAP18$_B$ ensembles closely resemble the apo-form, while all other HAP18-bound systems exhibit subtle curvature, demonstrating that this effect arises from C site occupancy and its minimal impact on neighboring B sites (Figs 5B,F and S3). Although the HAP1-bound system displays a broad, non-Gaussian distribution, perhaps owing to slower adaptation to B site vacancy, the ensemble shows evidence of converging to that of the HAP18-saturated state.

## CAMs alter interdimer interfaces even in unfilled sites

Despite the absence of CAMs in A sites, these interdimer interfaces register a response to occupancy in neighboring sites. Like B sites, quasi-equivalent A sites exhibit a minimal shift in $\theta_{spike}$, reflecting a preference for subtly curved conformations in the presence of CAMs. However, the interfacial adjustments appear agnostic to CAM class. The impact of AT130 is roughly the same as HAPs; however, a minor distinction between distributions is apparent depending on whether systems contain CAMs in B sites or are experiencing relaxation from the removal of CAMs from B sites (Figs 5C,G and S3). Since the contribution from filled B sites is small, the contribution from filled C sites is predominant. The latter observation is remarkable, since it demonstrates that structural consequences of C site occupancy can extend to A sites through B/D sites. Such dynamical coupling – propagating conformational changes up to two sites away – indicates an allosteric connection between interfacial pockets, even those that do not share a common chain. Considering the relative rigidity of the fivefold pentamer, [29] and the morphological constraints that govern icosahedral vertices, quasi-equivalent A sites may have the least freedom to adapt to local perturbations. Indeed, $\theta_{spike}$ for all CAM-affected capsids showed high mutual similarity (overlap >0.7, S3 Fig), suggesting limited pliability across the A-A interface.

While A sites are located in capsid pentamers, quasi-equivalent D sites are situated between B and C sites in capsid hexamers. Despite the absence of CAMs in D sites, these interdimer interfaces are subject to the structural consequences of B/C occupancy because they share a common chain with each neighboring site. Given that B sites undergo minimal adjustments to accommodate bound molecules, the impact on D sites is directly related to the extent of distortion induced in C sites upon CAM binding. Like A and B sites, quasi-equivalent D sites exhibit an increase in $\theta_{spike}$ in response to bound CAMs, but only in the presence of HAPs. The AT130-saturated capsid displays the same distribution as the apo-form, while all HAP-affected systems adopt more curved conformations across D sites (Fig 5D,G and S3). HAP system variants differentiate into two subtly different groups, this time depending on whether their C sites are filled or vacant. The high mutual similarity between $\theta_{spike}$ ensembles for groups with and without C site occupancy is compelling (overlap >0.9, S3 Fig) and further demonstrates dynamical coupling between C and D quasi-equivalent sites. The propagation of changes in one small molecule binding interface to another through the proteins that link them represents the basis for allostery. The responsiveness of D sites, and to a lesser extent A/B sites, to the state of C sites suggests that the latter is a central hub for allosteric influence within the intact capsid.

## CAM site accessibility depends on local occupancies

Shifts in interdimer angles affect the nature of dimer-dimer contact surfaces, with implications for CAM recognition. A previous study combining MD simulations and experiments revealed that hydrophobic contacts across the openings of CAM sites can partially occlude them, conferring some native drug resistance to pre-formed capsids [42]. These contacts occur especially between T109 of a given chain and V120 of the neighboring chain that caps it to form the interfacial pocket (Fig 7A, inset). Interestingly, the frequency of T109–V120 interaction was found to depend on quasi-equivalence, underscoring key differences between distinct dimer-dimer interfaces. In the absence of CAMs, pocket occlusion by T109–V120 is rare in B sites, but occurs >31% of the time in C sites (Fig 7A). This observation suggests that an isolated CAM is over 25× more likely to bind an apo-form capsid in a B site owing to its significantly higher rate of accessibility. However, while

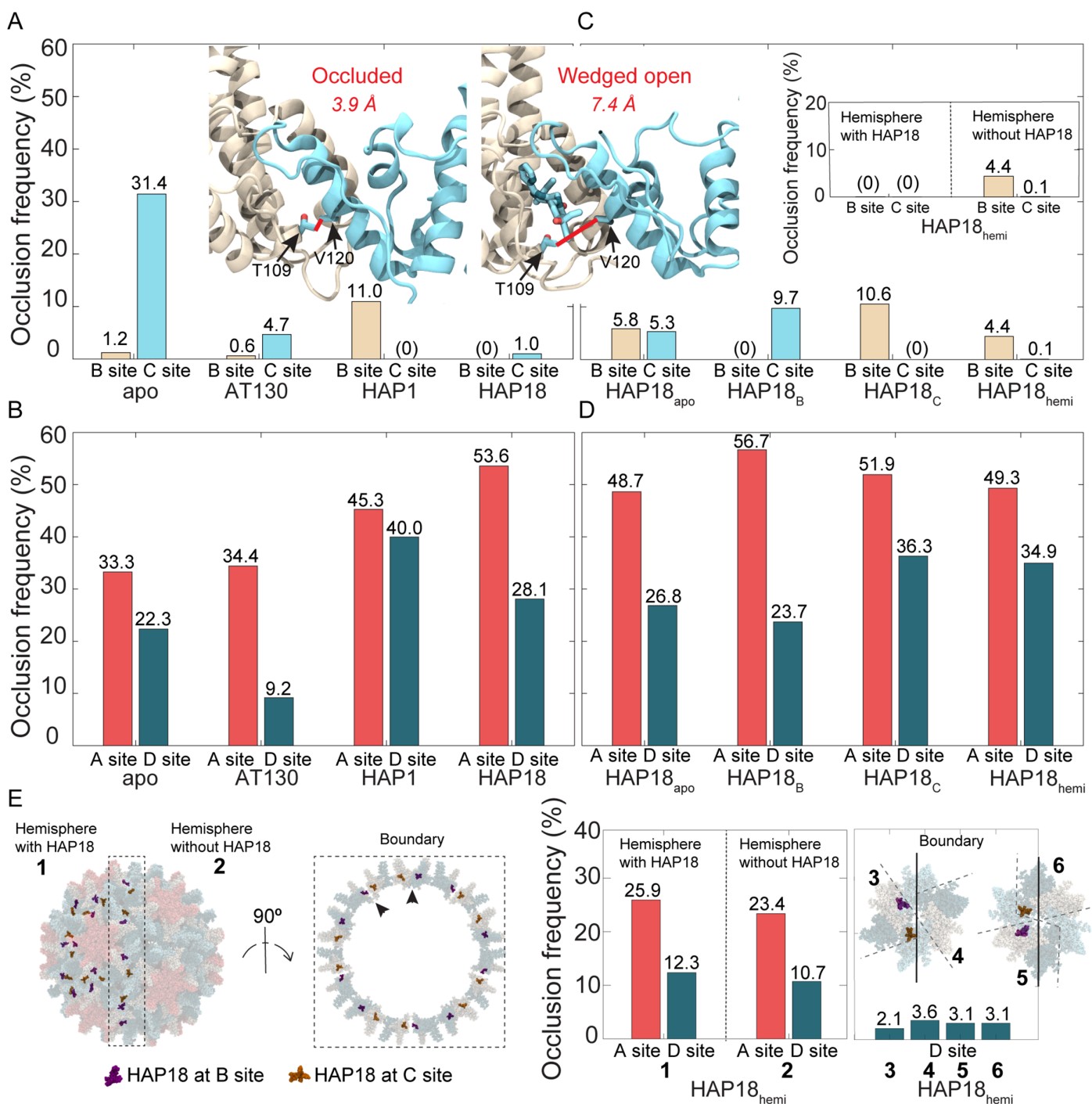

**Fig 7. T109–V120 contact frequency across CAM sites in simulated HBV capsids. (A, inset)** Hydrophobic interaction between threonine 109 and valine 120 of the adjacent chain reduces CAM site accessibility by occluding the entrance to the interdimer pocket [42]. Left illustrates inaccessible, vacant pocket that is occluded, while right illustrates accessible, CAM-occupied pocket that is unoccluded and wedged open. Contact between these residues is defined as $C\gamma$-$C\gamma$ distance <4.0 Å. Contact frequencies across **(A)** B/C sites and **(B)** A/D sites shown for apo-form, AT130-bound, HAP1-bound, HAP18-bound systems. Contact frequencies across **(C)** B/C sites and **(D)** A/D sites shown for $HAP18_{apo}$, $HAP18_B$, $HAP18_C$, $HAP18_{hemi}$ systems. **(C, inset)** Data for B/C sites in $HAP18_{hemi}$ decomposed by hemisphere. **(E)** D sites in $HAP18_{hemi}$ are characterized by six distinct local environments, depending on occupancy of neighboring B/C sites. Data for A/D sites in $HAP18_{hemi}$ decomposed by local environments. The sum of decompositions in panel **E** equals the bars shown in panel **D**. Frequencies are based on 1.5 million conformations sampled over the last 500 ns of simulation. Decompositions by hemisphere and local environments are normalized on the scale of full-capsid data for improved interpretation in S6 Fig.

PLOS Pathogens | https://doi.org/10.1371/journal.ppat.1013566    February 9, 2026

this naive probability may be relevant to uptake of the first CAM molecule, it does not account for complicating factors, such as induced fit or cooperativity in subsequent CAM binding.

Although some CAMs can simultaneously occupy all four quasi-equivalent sites in intact capsids (*e.g.,* DBT1 [26]), the subset examined here only exhibit B/C occupancy, [14–16] demonstrating that A/D occupancy is incompatible with the closed shell. Based on T109–V120 contact alone, MD simulations indicate that A/D sites are accessible nearly 70-80% of the time in apo-form capsids (Fig 7B), meaning additional structural characteristics contribute to their unsuitability for CAM recognition. Nevertheless, shifts in dimer-dimer contact surfaces signaled by T109–V120 frequency highlight the chemical-level consequences of interdimer adjustments between allosterically coupled sites. These local impacts on pocket topology and accessibility suggest that capsid dynamics – and their response to CAM occupancy – can affect the rate of subsequent CAM binding, and may explain cooperativity in CAM uptake reported by experiments [31].

T109–V120 hydrophobic contacts ($C\gamma$-$C\gamma$ distance <4 Å) are not captured for B/C sites in CAM-bound crystal structures (Table 2), yet they manifest periodically during MD simulations. Since capsids saturated with AT130 or HAP18 have their B/C sites wedged open by CAMs, they exhibit T109–V120 interaction less than 5% and 1% of the time, respectively (Fig 7A). The higher rate of occlusion in the former case arises from the greater adaptability of the AT130 binding mode (S5 Fig). Notably, in the AT130-bound system, the presence of CAMs in B/C sites has a negligible impact on A sites, but increases D site accessibility by 13% compared to the apo-form (Fig 7B). This observation is consistent with the propagation of C site distortion to D sites, although a meaningful change in $\theta_{\text{spike(D)}}$ was not observed (Fig 5D). In contrast, the presence of CAMs in B/C sites in the HAP18-bound system decreases A and D site accessibility, by 20% and 5% relative to the apo-form (Fig 7B), highlighting divergent effects of CAM-As versus CAM-Es on pre-formed capsid structure. In this case, C site distortion is distributed across the D site, through B and into A, while B site accessibility remains unchanged. Given that $\theta_{\text{spike(C)}}$ and $\theta_{\text{spike(D)}}$ indicate flattening of C sites concomitant with curving of B/A/D sites in response to HAP occupancy (Fig 5A-D), these data follow the anticipated trend that flat interdimer interfaces are more open and accessible compared to those that are curved.

### Neighboring CAM sites are allosteric and cooperative

The relaxation of previously CAM-saturated capsids following the removal of small molecules sheds further light on the coupling between quasi-equivalent binding sites. The HAP1-bound capsid exhibits no T109–V120 contact across its C sites, which are wedged open by CAMs; however, empty B sites, nearly always accessible in the apo-form capsid, are occluded about 10% more often when C sites are occupied (Fig 7A). The HAP18$_C$ system exhibits an equivalent response (Fig 7C), supporting the notion that B sites become less accessible upon filling of C sites. In contrast, the HAP18$_B$ system suggests that C sites become about 22% more accessible upon filling of B sites (Fig 7C). While these observations imply negative versus positive cooperativity, respectively, between the two quasi-equivalent sites, certainty in this conclusion is limited by the simulation timescale: It is unclear whether, over a longer period, HAP18$_B$ might ultimately relax to match the apo-form, which would mean that filled B sites have no impact on neighboring C sites. Indeed, behavior of the the HAP18$_{\text{apo}}$ system indicates caution in interpretation (Fig 7C), since its T109–V120 contact profile does not match

**Table 2**. T109-V120 distances in HBV capsid crystal structures.

| Crystal | PDB | A site† | B site† | C site† | D site† |
|---|---|---|---|---|---|
| apo-form | 2G33 | 3.93 | 6.08 | 3.77 | 4.46 |
| AT130-bound | 4G93 | 6.22 | 7.68 | 7.28 | 5.29 |
| HAP1-bound | 2G34 | 5.11 | 6.81 | 6.97 | 5.25 |
| HAP18-bound | 5D7Y | 3.49 | 5.55 | 6.34 | 3.50 |
| DBT1-bound | 6WFS | 4.11 | 4.99 | 5.24 | 4.23 |

†values given in Å.

that of the apo-form, demonstrating that the capsid has not yet recovered from the perturbation of total CAM removal. Still, HAP18$_{apo}$ exhibits a combined B/C occlusion frequency of about 10%, consistent with the individual values for empty C sites in HAP18$_B$ and empty B sites in HAP18$_C$ (Fig 7C). It may be, given that C sites require a more dramatic alteration to accommodate CAMs, that they are – in the reverse case – more immediately reactive to CAM removal; perhaps their spring-loaded relaxation to alleviate strain causes temporary B site distortion that will equilibrate out over the long term. This possibility would suggest that C sites have a faster relaxation rate than B sites.

Although both the HAP1-bound and HAP18$_C$ systems contain CAMs only in C sites and exhibit the most pronounced faceting, accessibility of neighboring A/D sites is reduced by differing extents (Fig 7D), perhaps owing to different small molecule sizes. These changes are 12% / 18% and 19% / 14% relative to the apo-form A/D sites, compared to 20% / 6% for the HAP18-saturated capsid. Thus, the unsubstituted HAP core is associated with a more modest local effect on A sites, despite inducing similar fivefold protrusion as the larger HAP. The additional presence of HAPs in B sites leads to less D site distortion than C site occupancy alone. The latter can be attributed to the observation that filled B sites result in more accessible C sites, such that C sites adjustments required to accommodate CAMs are minimized and reduced structural strain need be absorbed by D sites. As for the HAP18$_{apo}$ system, the behavior of its A/D sites follows a similar trend to other simulated capsids, in that they are significantly less accessible compared to B/C sites (Fig 7D). This outcome is consistent with the lack of CAM uptake in A/D sites for the small molecules under study in their analogous crystal complexes [14–16].

## CAM site distortion can propagate across the capsid

The HAP18$_{hemi}$ system reveals further insights into the distribution of strain across the capsid in response to bound CAMs. Like HAP18$_B$ and HAP18$_C$, this system contains 60 copies of HAP; however, the occupancy corresponds to one HAP-saturated hemisphere (30 each of filled B/C sites) and one apo hemisphere (30 each of empty B/C sites), rather than uniform distribution across the shell. T109–V120 contact frequencies indicate that C sites are almost always accessible, while B sites are rarely inaccessible, on par with their behavior in the HAP18$_{apo}$ case (Fig 7C). Most interesting is the observation that this pocket occlusion profile is representative of the hemisphere devoid of CAMs (Fig 7C, inset). The HAP18-saturated hemisphere behaves like the HAP18-saturated capsid, with all B/C sites wedged open by the bound molecules. Remarkably, the cumulative distortion appears to propagate across the entire capsid, retaining even empty C sites in the apo hemisphere as open and accessible. This response is in contrast to the relaxation of empty C sites observed in the HAP18$_B$ system, but may suggest that beyond some copy number threshold, the filling of C sites makes the filling of additional C sites more likely. The distortion induced by HAPs further propagates to empty B sites in the apo hemisphere, where the conformational adjustments that render C sites more accessible render their neighboring B sites less accessible, similar to the CAM site relationship observed in the HAP18$_C$ system. Thus, unlike the structural changes that increase capsid volume when both B and C sites are filled, those that alter CAM accessibility can have long-range impacts that accumulate and extend across the shell.

Considering the HAP18$_{hemi}$ results along with those for HAP1 and HAP18$_C$, it appears that C site occupancy in each half of the capsid (each 30 HAPs) induces approximately 5% B site occlusion, accounting for the values near 10% exhibited by the latter two systems (Figs 7A,C and S6A,B). An alternative viewpoint is that B site occlusion frequency for HAP18$_{hemi}$ is half that of the other two systems because half of the filled B sites are wedged open and protected from the distortions that would reduce their accessibility. Interestingly, A sites in HAP-saturated versus apo hemispheres mimic their counterparts in the HAP18$_C$ and HAP18$_{apo}$ systems, respectively (Figs 7D,E and S6C,D). However, the behavior of individual D sites depends on their local environment. Unlike A/B/C sites, which fall on either side of the boundary separating the two hemispheres, a subset of D sites rests on the boundary. Shell-wide, D sites are found in six distinct contexts (Fig 7E): within the *(i)* HAP-saturated or *(ii)* apo hemisphere, or at the boundary flanked by B/C sites that are *(iii)* both filled, *(iv)* both empty, *(v)* filled/empty, or *(vi)* empty/filled. While D sites in the first case are reminiscent of those in

HAP18$_C$, consistent with them absorbing distortion from adjacent occupied C sites, D sites in the other five cases exhibit varying responses (Figs 7D,E and S6C,D). When normalized on the scale of full-capsid data, it appears that type-*iii* D sites are not meaningfully perturbed by the occupancy of neighboring B/C sites, but that strain is instead propagated across the hemisphere boundary to the apo half of the hexamer, where type-*iv* D site occlusion is increased instead (S6D Fig). Type-*v* and *vi* D sites show an intermediate response, again similar to those in HAP18$_C$ (S6D Fig).

As with the HAP18$_{apo}$ case, it could be that the apo hemisphere of HAP18$_{hemi}$ has not fully recovered from the perturbation of widespread CAM removal. However, that the C sites of both hemispheres behave as if occupied over the course of simulation is at odds with the idea that they relax more rapidly that B sites. It may be that the filling of CAM sites beyond some copy number threshold leads to a collective response that slows the rate of local relaxation. Although the HAP18 capsids from which molecules were fully or partially removed exhibit long equilibration timescales in terms of volume (Fig 2B-C), suggesting that some structural perturbations remain unresolved, sphericity and faceting converge (Fig 2D-E), allowing for robust interpretation. A reasonable explanation for the propagation of C site adjustments across the entire capsid in HAP18$_{hemi}$ is the necessity for morphological match between the two hemispheres. HAP18$_B$ and HAP18$_C$ contain symmetric CAMs occupancy patterns, and their symmetric response to strain redistribution does not interfere with capsomer tiling suitable for a closed icosahedron. In contrast, faceting within the HAP-saturated hemisphere of HAP18$_{hemi}$ produces a decagonal bisection that distorts the more circular bisection of the apo hemisphere to maintain a closed interface between the two halves of the shell. Indeed, both the HAP18$_{hemi}$ and saturated HAP18-bound capsids display equivalent faceting angles (Fig 2F). Cause and effect can be reversed: While C site distortions induce faceting in the HAP-saturated hemisphere, the propagation of faceting induces C site distortions in the apo hemisphere. In the latter case, flattening of triangular faces leads to more open and accessible C sites regardless of CAM occupancy.

## Discussion

MD simulations of the HBV capsid offer a complementary perspective to crystallography and cryo-EM, which capture static, symmetrized structures that obscure the underlying conformational heterogeneity [29]. Experimentally-derived capsid models reflect averages across thousands of non-uniform particles, while MD demonstrates that – beyond the icosahedral architecture – symmetry is an ensemble property rather than a persistent feature of the intact shell. Capsids are instantaneously asymmetric, with local distortions tolerated through compensatory adjustments dynamically distributed across neighboring interfaces. Even flat subunit conformations seemingly incompatible with shell closure can be accommodated within the apo-capsid provided they are scattered events, whereas their widespread accumulation results in the faceted morphologies characteristic of CAM-A distortion that ultimately lead to rupture and dissociation. The interdimer interfaces themselves are variable, with native contact frequencies and response to CAM occupancy varying with quasi-equivalence. Overall, the all-atom view of capsid motion reveals the empty particle to be structurally plastic and adaptable, with the interleaved dimer arrangement representing a robust, flexibly-jointed framework.

Notably, experimental structures of CAM-bound capsids have been obtained using the Cp150 construct, a truncation mutant comprising the assembly domain with a C-terminal cysteine substitution at position 150 [38]. The engineered cysteine promotes disulfide cross-links between neighboring dimers, stabilizing the shell and protecting it from CAM-induced dissociation. This strategy allows structural characterization of complexes that would otherwise disassemble in the presence of CAM-As [14–16,26,31]. While indispensable for structure determination, Cp150 cross-links may constrain capsid dynamics, limit the flexibility of interdimer interfaces, or attenuate the extent of deformation that would occur in the absence of enforced stabilization. By contrast, MD simulations probe the unconstrained dynamics of Cp149, without cross-links and without imposed symmetry, allowing the observation of asymmetric distortions and lattice-level strain redistribution. The more pronounced deformations observed in HAP-bound capsids by MD relative to experimental structures may reflect both the absence of cross-linking and the fact that simulations explore physiological solution behavior, rather than conditions of crystal packing or vitrification. Indeed, the faceting and distortions apparent in MD likely

represent the early stages of destabilization, which on longer timescales culminate in disassembly and the formation of aberrant assembly products. Experimentally, capsid disruption in the presence of CAM-As proceeds over hours to days even at high small-molecule concentrations, [19] consistent with the metastable character of CAM-bound capsids observed here on the microsecond timescale.

The quasi-equivalent nature of interdimer interfaces in the HBV capsid provides distinct local environments that govern how CAMs interact with the shell and influence its morphology. MD simulations indicate that each quasi-equivalent site contributes differently to the likelihood of CAM binding and the structural consequences of bound CAMs, and the interplay of these sites establishes a framework for allosteric communication across the intact capsid lattice. A sites show the least structural plasticity among quasi-equivalent interfaces. Situated in pentamers, they are constrained by the geometric rigidity of the fivefold vertices and display limited conformational variability in response to CAM binding elsewhere. While CAM-induced orientational changes for dimers across A-A interfaces are modest, the impact on their contact surfaces is more pronounced. Shifts in A site accessibility occur via indirect coupling through neighboring B and D sites, although these adjustments are apparently insufficient to accommodate HAP or AT130 binding in pre-formed capsids. Importantly, A site responses indicate that even vacant and relatively inflexible sites remain linked to the allosteric network and participate in shell-wide strain redistribution. Biologically, the ability of the capsid to redistribute acute mechanical strain may be important for maturation or nuclear entry.

B sites appear intrinsically "CAM-ready." Among all quasi-equivalent interfaces, they appear most open and accessible. The AB-CD dimer pair that defines this site is incorporated into the lattice in a naturally flat conformation compared to unassembled intermediates, [39,41] effectively pre-organizing the site for CAM binding. The presence of HAPs in B sites induces only subtle conformational distortion, with little local strain and minimal impact on global morphology. Thus, uptake of CAM-As exclusively in B sites may not prompt pre-formed capsid dissociation, [43] and incorporation of HAP-bound dimers-of-dimers only as B sites might still permit formation of normal $T = 4$ capsids rather than misdirected assembly products. MD results suggest that the first CAM molecule is most likely to bind the intact capsid in a B site, and that B site occupancy can increase the accessibility of adjacent C sites, indicating potential positive cooperativity during CAM uptake. These observations provide a mechanistic explanation for cooperativity reported by spectroscopic studies of HAP-TAMRA binding to pre-formed capsids, [31] which along with cryo-EM data, imply the phenomenon arises from coupling between paired sites within hexamers.

C sites emerge as hubs of allosteric control within the capsid, as their occupation by HAPs exerts the most pronounced influence on local and global structure. They are accessible less often than B sites in the apo-form and undergo significant conformational distortion to accommodate CAM-As. These observations provide a mechanistic explanation for the results of kinetic studies, which indicate the rate of HAP-TAMRA uptake in pre-formed capsids is limited by a slow transition at the binding pocket [31]. Reshaping of C sites for CAM-A recognition may involve induced fit and produce the most consistent binding mode, given that HAP-bound structures show stronger small molecule density in C sites [14]. Filled C sites propagate structural changes to neighboring quasi-equivalent interfaces, decreasing the accessibility of both B and D sites. As such, C site occupancy may promote negative cooperativity by disfavoring subsequent binding at adjacent pockets. Cumulative local distortions have global effects: Filled C sites flatten CD-DC interfaces, induce hexamer bending, and drive faceting of the shell. The resulting morphological strain underlies the experimental observation that CAM-As destabilize pre-formed capsids [43]. Thus, C sites represent the primary lever for antiviral disruption of the intact shell and the critical target for drug design against core particles. Beyond some copy number, it appears that filled C sites increase accessibility of empty C sites as collective strain is broadcast across the assembly, suggesting that cooperativity also functions over long distances through the allosteric network.

D sites serve as structural sinks that absorb distortion induced by filled C sites. These interfaces are less accessible than B/C in the apo-form and are not observed to bind most CAMs experimentally, [14,15] yet their conformation responds strongly to C site occupancy. Indeed, HAP-bound crystal complexes show adjustments of both C and D chains, [16] consistent with participation of D sites in the conformational transition required for CAM uptake in C sites [31]. In particular,

D interfaces curve more dramatically when HAPs flatten neighboring C sites, exhibiting the most important compensatory motion that redistributes strain and preserves shell closure. D site accessibility is reduced by the presence of HAPs in C sites, but not in B sites, further emphasizing the coupling between C/D interfaces. By contrast, AT130 saturation increases the accessibility of D sites, while having no apparent effect on the orientation of the CD-BA dimer pairs that define them. This observation indicates caution in interpreting interfacial shits as solely rigid-body displacements, especially given that bound CAM-Es have been shown to induce significant tertiary structure changes, even as far away as the Cp spike [30]. Nevertheless, the responsiveness of D sites to HAPs highlights their role as secondary mediators of quasi-equivalent allostery.

Here, it is essential to distinguish between two complementary forms of inter-site communication. Formally, allostery refers to energetic coupling between binding sites, in which the complexation of a ligand in one site alters the conformational dynamics of a second, often distant site [44]. However, not all long-range structural communication qualifies as allosteric in this sense. A related phenomena is tensegrity ("tensional integrity"), [45] which involves mechanical coupling between sites: Local strain can be redistributed within a closed geometric shell, allowing systems like virus capsids to withstand acute structural deformation, while simultaneously communicating conformational changes from one location to another [46]. The HBV capsid clearly exhibits tensegrity properties, including in response to CAM binding, [31] and the propagation of strain induced by CAM occupancy represents a major mechanism underlying communication between quasi-equivalent interfaces. Given that CAM binding sites display cooperativity, the tensegrity effect functions practically as allostery with respect to small molecule uptake in the intact capsid. The Cp dimer also appears to be allosteric in the traditional sense, exhibiting energetic coupling between the inter- and intradimer interfaces [12,30,47,48]. In that case, tensegrity and allostery likely coexist in the HBV system, and may jointly mediate the structural effects of CAM binding on capsid assembly and stability. The observation that CAMs can exploit these mechanisms of conformational control over the capsid led to their early designation as core protein allosteric modulators (CpAMs), prior to revision of the nomenclature [49].

The CAMs examined here highlight the divergent consequences of occupying B and C sites in pre-formed capsids. HAP molecules, as CAM-As, accelerate assembly but also misdirect it, producing aberrant products and destabilizing intact capsids [18,19]. MD simulations indicate that HAPs "overfill" B and particularly C interfaces, such that accumulated strain propagates across the lattice and primes the shell for rupture. Notably, occupancy of both B and C sites increases capsid volume regardless of CAM chemotype, whereas occupancy of B or C alone does not. Size dilation is observed via coarse-grain and atomistic descriptions (Figs 2A-B and S7), confirming that it is not an artifact of the polyhedral abstraction. Yet volume increase alone does not dictate morphological outcome: The most spherical and most faceted capsids, apo-form and HAP1-bound (in C sites only), exhibit the same volume state. Thus, while saturation of B/C sites with HAPs is sufficient to destabilize the shell, the underlying mechanism involves local deformation and redistribution of strain rather than simple volumetric expansion. In contrast, AT130, a CAM-E, also occupies B/C sites and induces similar volume increases to HAPs, yet capsids assembled in its presence remain morphologically normal. AT130 appears to make D sites more accessible, while altering tertiary structure, [30] such that local strain redistribution occurs in a manner that avoids global disruption. A key difference is the high adaptability of the AT130 binding mode (S5 Fig), which is also apparent in free dimers-of-dimers [39,50]. These results underscore that the number of CAMs bound per capsid does not correlate directly with disruptive potential. Instead, distinct binding modes of different chemotypes and their associated mechanism of strain propagation confer divergent consequences for intact capsid stability.

DBT1 presents a further contrast, as this CAM-A has been shown to bind all four quasi-equivalent sites in intact capsids, including A/D. While the only other examples of truly CAM-saturated structures are aberrant assemblies or hexamer sheets, [51,52] DBT1's unique binding mode allows it to simultaneously fill all 240 interdimer interfaces in the $T = 4$ shell, given Cp150 cross-links [26]. Consistent with the MD-based observation that filling additional quasi-equivalent positions leads to volumetric expansion of the capsid, the DBT1-bound cryo-EM structure exhibits larger volume than capsids with only B/C site occupancy (Fig 2A, triangular point) [26]. Considering this trend, it is possible that the "third size state"

initially sampled by the HAP18-bound system during simulation reflects occasional A/D occupancy that was insufficiently frequent to be resolved by cryo-EM [16]. Although T109–V120 contacts in the HAP18 crystal contrarily suggest A/D site occlusion (Table 2), the interdimer contacts captured in static structures are not always representative of these highly dynamic interfaces [42]. Interestingly, the DBT1 complex also appears more spherical than other CAM-bound capsid models (Fig 2D, triangular point), suggesting that local distortions are propagated across the shell via an alternative mechanism than with HAPs or AT130. Altogether, these observations highlight that (i) binding mode, (ii) quasi-equivalent occupancy, (iii) saturation level, (iv) Cp allostery, and (v) tensegrity response all play a role in determining capsid stability in the presence of different CAM chemotypes and whether their binding will lead to normal morphologies, faceting, or induced disassembly.

The ability of CAMs to destabilize pre-formed capsids is of particular therapeutic relevance, as it provides a mechanism to impair genome-containing core particles that are critical to the HBV life cycle. While RNA-filled (immature) nucleocapsids exhibit enhanced mechanical stability due to counterbalancing interactions with the CTD and the enclosed pregenome, [52–54] and thus resist CAM-induced disruption, DNA-filled (mature) nucleocapsids are comparatively unstable and under internal stress that renders them more susceptible to morphological perturbation [6,55–57]. CAM-induced strain can therefore sensitize mature capsids to premature uncoating, blocking the delivery of the relaxed circular (rcDNA) to the nucleus and decreasing the formation of covalently closed circular DNA (cccDNA) [24,58]. Importantly, cccDNA serves as the replication template, and chronic HBV infection is characterized by a persistent nuclear reservoir of cccDNA that drives continuous viral production [3,59]. Although current therapies (i.e., nucelos(t)ide analogs) block reverse transcription to suppress replication, they do not eliminate the cccDNA, leading to relapse post treatment [2]. Antiviral strategies that limit reinfection and deplete the cccDNA reservoir have the potential to diminish the persistence of HBV in hepatocytes, and thus represent one of the most promising approaches to achieving a functional cure for chronic HBV infection [60–62].

## Materials and methods

### CAM parameterization

Parameterization of HAP1 and AT130 molecules was previously described [28,30]. Custom force field parameters compatible with CHARMM36 [63] were developed here for HAP18. The CHARMM general force field (CGenFF) program [64] was used to assign atom types and partial atomic charges by analogy, as well as to identify missing parameters not covered by CGenFF. The missing bond, angle, and dihedral parameters, CGenFF parameters with high penalty scores, and the partial charges of all atoms were optimized using standard protocols in the Force Field Toolkit (ffTK) [65] plugin in VMD [66]. Quantum mechanics (QM) calculations were performed with Gaussian09 [67]. The 63-atom HAP18 molecule was reduced to a 28-atom fragment (Fig 8A) to isolate key parameters requiring derivation/refinement and reduce the computational overhead of QM calculations. The final potential energy profiles of refitted dihedral parameters demonstrated excellent agreement with those computed with QM (Fig 8B). CGenFF and ffTK/QM-derived parameters/charges were combined to yield a complete force field description for HAP18.

### Computational modeling

Computational modeling to rebuild unresolved C-terminal residues in the Cp149 capsid in its apo-form and bound to AT130 was previously described [29,30]. An equivalent protocol was applied here to construct complete models of the Cp149 capsid bound to HAP1 and HAP18, based on available crystal structures of their complexes [14,16]. Using ROSETTA, [68] missing residues in the capsid pentamer (five A chains) were folded simultaneously to generate 2,000 structures. Missing residues in capsid hexamer (two B chains, two C chains, and two D chains) were folded simultaneously to generate 5,000 structures. Besides the two residues preceding the modeled regions, crystallographically-resolved residues were held fixed. In total, an ensemble of 10,000 structural models for each chain were produced. The ensembles

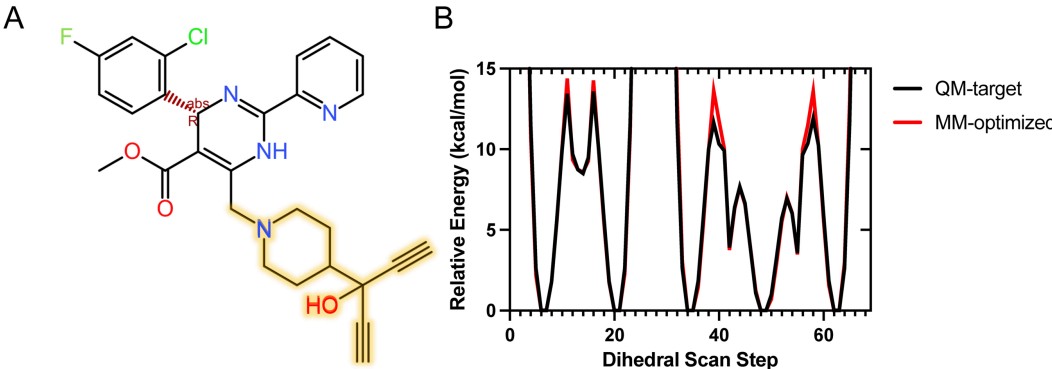

**Fig 8. Derivation of classical force field parameters for HAP18. (A)** HAP18 was reduced to a smaller fragment, highlighted in yellow, for QM calculations. This region was not well covered by CGenFF, requiring additional parameter derivation and refinement. **(B)** Optimized dihedral parameters show accurate reproduction of QM rotational profiles.

were clustered based on protein backbone RMSD, using a partitioning around medoids (PAM) scheme, as implemented previously [69]. The medoids for the most populated cluster for each quasi-equivalent chain were selected as likely conformations and integrated into the crystal asymmetric unit. Cispeptides in the HAP18 crystal structure were corrected using the cispeptides plugin in VMD [66]. Final capsid models were produced by applying the transformation matrices provided in the respective PDBs. Protonation states of titratable groups and all other hydrogen coordinates were assigned to the capsids for pH 7.0 using PDB2PQR [70]. Sodium and chloride ions were placed around each capsid to neutralize the local electrostatic potential using the CIonize plugin in VMD [66]. Each capsid with local ions was solvated in a 39.2 nm$^3$ box of TIP3P water [71] containing 150 mM NaCl. Simulation files were prepared using the psfgen plugin in VMD, [66] applying the CHARMM36 force field [63]. Capsid systems including explicit solvent contain approximately six million atoms.

## Molecular dynamics simulations

All-atom MD simulations were performed with NAMD 2.13 [72] on the Blue Waters supercomputer. The energy of each system was minimized using the steepest descent algorithm for 30,000 cycles for the solvent, then for the solvent and the protein side chains. Subsequently, systems were heated in the NPT ensemble, increasing the temperature from 60 K to 310 K over 5 ns, applying Cartesian restraints of 5 kcal/mol to the protein backbone. After reaching a temperature of 310 K, backbone restraints were removed gradually over an interval of 5 ns. Following 5 ns of equilibration, 1 $\mu$s sampling was collected using a timestep of 2 fs, saving coordinate frames every 20 ps. The r-RESPA integrator was used to propagate dynamics. Electrostatic interactions were split between short and long range at a cutoff of 12 Å according to a quintic polynomial splitting function. Long-range electrostatics were computed with the particle-mesh Ewald (PME) method using a grid spacing of 2.1Å and 8$^{th}$ order interpolation. Full electrostatic evaluations were performed every 4 fs. The temperature was regulated with the Langevin thermostat, employing a damping coefficient of 1 ps$^{-1}$. The Langevin piston Nosé-Hoover method was applied to maintain a constant pressure of 1.0 bar at isotropic conditions, with a piston oscillation period of 2000 fs and a damping timescale of 1000 fs. Bonds to hydrogen atoms were constrained with the SHAKE algorithm for solute and the SETTLE algorithm for solvent. The final 500 ns of conformational sampling (25,000 frames) was used for analysis. Analogous sampling was extracted from previously published trajectories of the apo-form and AT130-bound capsids, [29,30] which were collected with identical simulation engine, force field, and protocol.

## Trajectory analysis

The reversible response of the capsid to the presence or removal of bound CAMs was characterized in terms of local and global structural perturbations, as described below, rather than measuring elastic strain directly. Thus, the term "strain" is used in this work in a qualitative sense. MD trajectory analysis was carried out with VMD 1.9.4 [66].

## Shell volume and sphericity

Shell volume and sphericity were determined by fitting a polyhedron of 240 triangular faces to the capsid surface, as previously described [29]. Vertices were defined as the geometric centers of capsid pentamers (fivefolds), hexamers (quasi-sixfolds), triangular pores (threefold interfaces between hexamers), and psuedo-triangular pores (pseudo-threefold interfaces between two hexamers and one pentamer). This partitioning is equivalent to dividing each pentagon into five triangles and each hexagon into six triangles. Vertex definitions included $C\alpha$ atoms of the three to six chains comprising each subunit. The volume of the polyhedron $V$ was calculated by summing the volumes of 240 triangular pyramids – each formed by a triangular face and the center of the capsid – according to Eq 1, where $b_i h_i$ is the area of the base of a triangular pyramid $i$ and $H_i$ is the pyramid height. Capsid sphericity $\Psi$ was calculated as the ratio of the volume of the polyhedron to its surface area, according to Eq 2.

$$V = \sum_{i=1}^{240} \frac{(b_i h_i / 2) H_i}{3} \tag{1}$$

$$\Psi = \frac{\pi^{1/3}(6V)^{2/3}}{\sum_{i=1}^{240} b_i h_i / 2} \tag{2}$$

## Shell faceting

Faceting was characterized by fitting a polyhedron of 60 kite-shaped faces to the capsid surface, as previously described [28]. Each kite, defined by the pentamer, hexamer, and triangular-pore vertices from above, represents an asymmetric unit. The plane normal of each kite relative to local fivefold, quasi-sixfold, and threefold symmetry axes yields the angles $\varphi_5$, $\varphi_6$, and $\varphi_3$, which quantify the inclination of the faces around each vertex. Increases in angles indicate protrusion of the shell, while decreases indicate flattening. The relative protrusion of kites around fivefolds (the icosahedral vertices) versus flattening around threefolds (the icosahedral faces), captured by $\varphi_5 - \varphi_3$, yields the so-called faceting angle. Values are averaged over 60 symmetric copies.

## Interdimer angles

Each quasi-equivalent dimer was represented by two vectors, as previously defined, [39] based on the geometric centers of specified atom clusters. The first vector, corresponding to the direction of the Cp *spike*, was fit to the chassis region of the four-helix bundle, along the intradimer interface. The initial point was assigned using the backbone atoms of residues 49-56 and 103-110. The terminal point was assigned using the backbone atoms of residues 56-65 and 96-103. The second vector, corresponding to the perpendicular direction, was fit across the *base* of the dimer, spanning both copies of helix-5. The initial and terminal points were assigned using the backbone atoms of residues 111-127 for each chain, respectively. Relative orientations of paired dimers across capsid hexamers were calculated as the angles between spike vectors, as illustrated in Fig 3A, denoted $\phi_{BA}$ for the AB-BA pair, $\phi_{CD}$ for the DC-CD pair, and $\phi_{DC}$ for the CD-DC pair. Relative orientations of paired dimers across interdimer interfaces were calculated as the angles between spike vectors, $\theta_{spike}$, and the angles between base vectors, $\theta_{base}$, as previously described [39]. Distribution overlap scores were

determined based on the shared area between normalized histograms using the Histogram Intersection Similarity Method (HISM) [40].

## Solvent exchange rates

Solvent exchange rates were calculated as previously described for the capsid, [29] based on the number of water molecules (or ions) that were found on one side of the shell (inside/outside) at a reference time $t_0$ and then on the opposite side (outside/inside) at time $t_0 + \Delta t$. The interior versus exterior of the capsid was classified using the GPU implementation of VMD's *measure volinterior* [73] function with fixed-boundary detection. Parameters were selected empirically to produce a closed surface representation (*Radius Scale* 3.6, *Isovalue* 1.8, *Grid Spacing* 1.0 Å), and 64 rays were cast per frame. Exchange events were counted every 5 ns over 25-ns intervals before resetting the reference frame. Rates were determined from the slope of the linear fit to the number of exchange events plotted versus simulation time (S8 and S9 Figs). Average rates are reported over 1,000 slopes, representing the last 500 ns of conformational sampling.

## Supporting information

**S1 Fig. Statistical analysis of sphericity values.** (**A**) Sphericity values were extracted from the last 500 ns of each simulation to minimize the influence of initial equilibration. Histograms of these values show minimal overlap between the apo-form and CAM-bound systems, with the apo-form displaying sphericity values not sampled among the other capsid simulations. To determine the appropriate statistical test for comparing sphericity across capsid systems, the normality of these distributions was assessed using the D'Agostino-Pearson test, which evaluates deviations in skewness and kurtosis from a normal distribution. The null hypothesis ($H_0$) that each dataset follows a normal distribution was rejected for all systems (p-values $\ll 1 X 10^{-8}$). (**B**) Differences among sphericity distributions were evaluated using the Kruskal-Wallis test, a non-parametric test with the $H_0$ that all sphericity distributions are statistically indistinguishable. Given an extremely small p-value (p-values $\ll 1 X 10^{-8}$), $H_0$ was rejected meaning at least one sphericity distribution is statistically different. Pairwise comparisons were then performed using Dunn's post-hoc test with Bonferroni correction, in which small absolute z-scores and adjusted p-values close to unity indicate that only the HAP1-bound and HAP18$_C$ sphericity distributions are statistically indistinguishable. All other pairwise comparisons show statistically significant differences between distributions. (TIFF)

**S2 Fig. Average solvent exchange rates.** Inward (*In*) and outward (*Out*) exchange rates over the last 500 ns are shown for (**A,C,E**) apo-form, AT130-bound, HAP1-bound, and HAP18-bound systems, and (**B,D,F**) HAP18$_{apo}$, HAP18$_B$, HAP18$_C$, HAP18$_{hemi}$ systems. Panels show (**A-B**) water, (**C-D**) sodium, and (**E-F**) chloride exchange rates. Black dots denote the mean between inward and outward rates per system; error bars represent standard deviations. (TIFF)

**S3 Fig. Overlap scores between interdimer angle distributions.** Scores are shown for each quasi-equivalent CAM-binding site: (**A**) C site, (**B**) B site, (**C**) A site, and (**D**) D site. Left panels show overlaps for $\theta_{spike}$ distributions, while right panels show overlaps for $\theta_{base}$ distributions. Pairwise overlap scores are reported for all system combinations, where values near 0 indicate minimal similarity and values near 1 indicate high similarity between distributions. (TIFF)

**S4 Fig. Standard deviation ellipses (SDEs) for interdimer angles.** SDEs are shown for (**A**) C site, (**B**) B site, (**C**) A site, and (**D**) D site. Left panels correspond to apo-form, AT130-bound, HAP1-bound, and HAP18-bound systems, while the right panels show HAP18$_{apo}$, HAP18$_B$, HAP18$_C$, and HAP18$_{hemi}$ systems. Each ellipse plots the relationship between $\theta_{spike}$ (y-axis) and $\theta_{base}$ (x-axis), centered on the mean values. Ellipse radii reflect the standard deviation of each angle, and orientation is determined by the covariance between $\theta_{spike}$ and $\theta_{base}$. A 90% confidence level was used. (TIFF)

**S5 Fig. Adaptability of the AT130 binding mode.** Extracted conformations from the AT130-bound capsid exhibit hydrophobic occlusion between T109 and V120, yet the AT130 molecule is still accommodated through distinct binding modes. Side chains of T109 and V120 are indicated with black arrows. The inter-residue distances are indicated above each conformation.
(TIFF)

**S6 Fig. Alternative normalization of HAP18$_{hemi}$ T109-V120 contacts.** Contact frequencies across (**A,B**) B/C sites, and (**C,D**) A/D sites. Occlusion frequencies in (**A,C**) are normalized over 1.5 million conformations sampled over the last 500 ns of simulation, reflecting the cumulative sampling of 60 copies of each CAM site in a capsid conformation. Frequencies in (**B,D**) are recalculated by normalizing only over the conformations belonging to the same local environment, which improves direct comparison with full-capsid data. A, B, and C sites each display two local environments, corresponding to 30 of the 60 copies per capsid (750,000 conformations). D sites exhibit six local environments due to their boundary location. D sites at the hemispheres correspond to 20 of the 60 copies (500,000 conformations), while the four boundary environments correspond to 5 copies each (125,000 conformations per environment).
(TIFF)

**S7 Fig. Radius of gyration (Rg) of simulated HBV capsids.** Rg over time for (**A**) apo-form, AT130-bound, HAP1-bound, HAP18-bound systems. Scatter points on the left indicate Rg values measured from the corresponding crystal structures.Time evolution of Rg for (**B**) HAP18$_{apo}$, HAP18$_B$, HAP18$_C$, and HAP18$_{hemi}$ systems.
(TIFF)

**S8 Fig. Solvent exchange rates over time for apo-form, AT130-bound, HAP1-bound, and HAP18-bound capsids.** Number of (**A**) water molecules, (**B**) sodium ions, and (**C**) chloride ions. Exchange rates were calculated as the slopes of linear regressions fitted to data sampled every 5 ns within 25 ns intervals, with final rates taken as the average of all slopes. Inward and outward exchanges are shown in red and blue, respectively. Analyses were performed over the last 500 ns of each simulation.
(TIFF)

**S9 Fig. Solvent exchange rates over time for HAP18$_{apo}$, HAP18$_B$, HAP18$_C$, and HAP18$_{hemi}$ capsids.** Number of (**A**) water molecules, (**B**) sodium ions, and (**C**) chloride ions. Exchange rates were calculated as the slopes of linear regressions fitted to data sampled every 5 ns within 25 ns intervals, with final rates taken as the average of all slopes. Inward and outward exchanges are shown in red and blue, respectively. Analyses were performed over the last 500 ns of each simulation.
(TIFF)

## Acknowledgments

Coordinates for HAP1-bound HBV capsid structure kindly provided by Professor Adam Zlotnick, Indiana University. This research was part of the Blue Waters sustained-petascale computing project, supported by the National Science Foundation (NSF, awards OCI-0725070 and ACI-1238993) and the state of Illinois. Blue Waters was a joint effort of the University of Illinois at Urbana-Champaign and its National Center for Supercomputing Applications. Computer time for MD simulations was provided by the Great Lakes Consortium for Petascale Computation for project: "Furthering characterization of the hepatitis B virus capsid as a drug target: Simulations to investigate quasi-equivalence and cooperativity in drug binding." Computer time for MD trajectory analysis on Delta at University of Illinois at Urbana-Champaign and Anvil at Purdue University was provided by allocations BIO230131 and BIO240029 through the Advanced Cyberinfrastructure Coordination Ecosystem: Services & Support (ACCESS) program (NSF awards #2138259, #2138286, #2138307, #2137603, and #2138296). This research was also supported by the Delaware Advanced Research Workforce and Innovation Network

(DARWIN, NSF award OAC-1919839) and the BioStore resource made possible by the National Institutes of Health (NIH, awards P20-GM-103446 and S10-OD-028725).

## Author contributions

**Conceptualization:** Jodi A. Hadden-Parilla.

**Data curation:** Jodi A. Hadden-Parilla.

**Formal analysis:** Carolina Pérez-Segura, Jodi A. Hadden-Parilla.

**Funding acquisition:** Jodi A. Hadden-Parilla.

**Investigation:** Carolina Pérez-Segura, Jodi A. Hadden-Parilla.

**Methodology:** Carolina Pérez-Segura, Boon Chong Goh, Jodi A. Hadden-Parilla.

**Project administration:** Jodi A. Hadden-Parilla.

**Resources:** Jodi A. Hadden-Parilla.

**Software:** Carolina Pérez-Segura, Jodi A. Hadden-Parilla.

**Supervision:** Jodi A. Hadden-Parilla.

**Validation:** Carolina Pérez-Segura, Jodi A. Hadden-Parilla.

**Visualization:** Carolina Pérez-Segura, Jodi A. Hadden-Parilla.

**Writing – original draft:** Carolina Pérez-Segura, Jodi A. Hadden-Parilla.

**Writing – review & editing:** Carolina Pérez-Segura, Jodi A. Hadden-Parilla.

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
