## [Decision Letter · Decision Letter 0]

7 Nov 2025

PPATHOGENS-D-25-02380

Mechanistic insights into CAM-induced disruption of HBV capsids revealed by all-atom MD simulations

PLOS Pathogens

Dear Dr. Hadden,

Thank you for submitting your manuscript to PLOS Pathogens. After careful consideration, we feel that it has merit but does not fully meet PLOS Pathogens's publication criteria as it currently stands. Therefore, we invite you to submit a revised version of the manuscript that addresses the points raised during the review process.

We look forward to receiving your revised manuscript.

Kind regards,

Michael D Robek

Academic Editor

PLOS Pathogens

Robert Kalejta

Section Editor

PLOS Pathogens

Sumita Bhaduri-McIntosh

Editor-in-Chief

PLOS Pathogens

orcid.org/0000-0003-2946-9497

Michael Malim

Editor-in-Chief

PLOS Pathogens

orcid.org/0000-0002-7699-2064

**Journal Requirements:**

At this stage, the following Authors/Authors require contributions: Carolina Pérez-Segura, Boon Chong Goh, and Jodi Hadden-Perilla. Please ensure that the full contributions of each author are acknowledged in the "Add/Edit/Remove Authors" section of our submission form.

**Reviewers' Comments:**

Reviewer's Responses to Questions

**Part I - Summary**

Reviewer #1: The authors employ extensive all-atom molecular dynamics simulations to explore how different capsid assembly modulators (CAMs) influence the structure and mechanical response of the hepatitis B virus capsid. The authors show that drug binding at specific interdimer interfaces transmits local strain through the lattice, producing distinct effects for each CAM class: CAM-A molecules (HAP1, HAP18) generate lattice distortion and destabilization, whereas the CAM-E molecule (AT130) preserves capsid geometry while locking it into a functionally inert state. The analysis also identifies B and C sites as central nodes of allosteric communication and offers a structural rationale for the cooperative drug-binding behavior reported experimentally.

Overall, the study provides a convincing atomistic explanation for how small molecules can either disrupt or over-stabilize the HBV capsid and contributes valuable mechanistic insight into antiviral strategies targeting core–protein interactions. I think that this is an interesting paper and I recommend it for publication, but only after the authors respond to my concerns raised below.

Reviewer #2: The paper from Perez-Segura et al. describes microsecond atomistic simulations of HBV capsids with various capsid assembly modulators (CAMs) bound. The authors find that different interfaces have distinct responses. The coupling between the interfaces redistributes the strain in such a way as to create "functionally allosteric" effects. These observations explain the known effects of different CAMs.

This is an impressive study. I have a few questions and suggestions for improvement.

1. On page 4 the authors discuss how long it takes for the capsid volume to stabilize. I would argue that it is difficult to determine that if only one simulation per system was performed. How confident are the authors that similar results would observed if the simulations were repeated?

2. Figure 2 should be made clearer; see specific points a-d below.

a. I assume that all colors are the same as labeled in the F subplot? This should be explicitly stated in the caption.

b. For Figure 2A, I guess the x10^7 applies to the y-axis, but its positioning is less than ideal.

c. For Figure 2D, the authors show a consistent decrease in sphericity for all CAM-bound capsids; however the difference is very small in comparison to the apo capsid, between 0.2-0.4%. How can we be sure that these differences are statistically significant?

d. For Figure 2E, it would be very helpful to have an SI figure that illustrates the three angles measured here.

3. On line 152, the subsection headline is "CAMs promote hexamer bending and asymmetry". While this seems to be true for CAM-A based on the presented data, the CAM-E results or AT130 are not discussed much in this subsection, and the data in Figure 3F shows the angle distributions to be very similar to the apo case. Therefore, would "CAM-As promote hexamer bending and asymmetry" be a better title and more representative of the described data?

4. Figure 7A and C, the authors should clearly label residues 124 and 120 here.

5. In discussion lines 670-672 the authors state: "Notably, occupancy of both B and C sites increases capsid volume regardless of CAM chemotype, whereas occupancy of B or C alone does not."

I would argue that the paper only shows this for HAP18 and not generally for all CAMs. HAP1 was only simulated with C occupancy, and AT130 with both C and B, so we don't know if they would show the same trends as HAP18 for different occupancy choices. Also the HAP18 simulations were only performed once, so consistency of this result was not shown. The authors should add that this was only shown for HAP18 in the paper.

6. In methods lines 784-785 the authors state that "Analogous sampling was extracted from previously published trajectories of the apo-form and AT130-bound capsids."

Could the authors specify if there were any differences between these older simulations and the new ones with regards to force field and MD protocol?

Reviewer #3: This manuscript reports the results from all-atom molecular dynamics (MD) simulation studies of CAM binding of HBV capsids derived from Cp149. The study reveals the mechanism of CAM binding at the “HAP” pocket between Cp dimer-dimer interfaces to induce the structure changes and disassembly of intact capsids. Interestingly, the study revealed that each quasi-equivalent interface exhibits a unique response. While the A and D sites are unfilled, B sites are the most open and C sites are the hubs of allosteric control and the key CAM target, as their occupancy creates local distortion that is broadcast to adjacent sites, driving capsid faceting and destabilization. Interestingly, the extent of C site adjustment and the nature of D site counterbalance vary with CAM chemotypes and underlines the distinct phenotypes of CAM-A and CAM-E on assembled capsids and highlights the allosteric (or tensegrity) nature of CAM action. The findings of this study shed light on the development of novel CAMs that can more efficiently induce the disassembly of nucleocapsids to inhibit HBV cccDNA synthesis.

Overall, the study is well conceived and executed. The results are clearly presented. The findings are of significance to understand CAM antiviral mechanisms and development of next generations of CAMs with improved pharmacological properties.

**Part II – Major Issues: Key Experiments Required for Acceptance**

Reviewer #1: N/A

Reviewer #2: (No Response)

Reviewer #3: Despite briefly discussed in the manuscript, it will be very interesting to determine the dynamic structure alteration and disassembly of capsids induced by dibenzothiazepines (DBTs) by all-atom molecular dynamics (MD) simulations.

**Part III – Minor Issues: Editorial and Data Presentation Modifications**

Reviewer #1: 1. The abstract includes too much technical detail, especially about quasi-equivalent sites. For readers who are not familiar with this concept, the main idea is hard to follow. The abstract would be clearer if it focused on the overall findings and biological relevance instead of structural specifics.

2. The discussion of CAM-E molecules is confusing. They are described as accelerating assembly and producing morphologically normal but empty capsids, but the simulations use Cp149, which lacks the C-terminal domain and cannot encapsidate the genome. It is unclear whether the experimental data cited involve full-length core protein or genome-containing capsids. Please clarify this point.

3. Can the authors compare and contrast their results with those of Tresset’s group (ACS Nano 17, 12723–12733, 2023), which experimentally studied the energetics and kinetic assembly pathways of HBV capsids in the presence of CAMs?

4. Please clarify what is meant by the apo form. Does this refer to capsids without any bound CAM molecules, and were these systems equilibrated under the same simulation conditions as the CAM-bound ones?

5. It would be helpful to include representative structural snapshots for each case in Fig. 2 to visualize the morphological differences under various CAM-binding conditions.

6. The description of CAM-bound capsids as “more faceted” is not clear. Does this mean the particles show stronger icosahedral edges and faces, or that they deviate from icosahedral symmetry?

7. The statement that “in the T = 4 capsid, hexamers are positioned on the icosahedral edges” is ambiguous. Does this mean the centers of hexamers lie along the edges connecting adjacent pentamers?

8. The angular parameters ϕ₅, ϕ₃, and ϕ₆ need to be defined more clearly, including the directions in which they are measured relative to the capsid center. A simple schematic showing how the faceting angle (ϕ₅ − ϕ₃) is determined would make this section easier to follow.

9. The manuscript repeatedly refers to “allosteric control” and mentions that the dimer is “allosteric in the traditional sense.” Please clarify what kind of allostery is meant—mechanical strain propagation across the lattice, conformational coupling within dimers, or both—and what evidence from the simulations supports this claim. Recent MD simulations (Science Advances 11, e ady7241, 2025) showed that allosteric coupling between capsid proteins promotes lattice elasticity but that complete T = 4 shells form only in the presence of the genome. It would help to comment on how genome-free Cp149 simulations relate to the allosteric and assembly behavior of genome-containing capsids.

10. The manuscript states that “HAP molecules, as CAM-As, accelerate assembly but also misdirect it,” but CAM-As are known to misdirect or destabilize assembly rather than accelerate it. In contrast, CAM-E molecules accelerate assembly and lead to empty but morphologically normal capsids. Since the simulations examine only pre-formed capsids, they cannot directly address assembly kinetics or misdirection. Please clarify that the results concern structural responses of mature capsids to CAM binding and separate the simulation findings from experimental conclusions.

11. The paper frequently refers to strain propagation and redistribution, but the terms “strain” and “stress” are not clearly defined. See, for example, ACS Nano 16, 317 (2022) for the experimental definition of stress in Dragnea’s group’s work. It would be helpful if the authors clarified whether these terms are used qualitatively or in a quantitative mechanical sense, and whether they correspond to measurable quantities such as local elastic strain or interfacial deformation. A brief clarification in the Methods or Discussion would make the tensegrity model easier to interpret.

Reviewer #2: (No Response)

Reviewer #3: Line 36, because HBV is not a naked capsid virus, but an enveloped virus, the statement that “… leading to the production of non-infectious particles” is not accurate. It should be “…leading to the production of aberrant structures or empty capsids”.

PLOS authors have the option to publish the peer review history of their article (what does this mean?). If published, this will include your full peer review and any attached files.

Reviewer #1: No

Reviewer #2: No

Reviewer #3: No

**Figure resubmission:**
---

## [Decision Letter · Decision Letter 1]

22 Jan 2026

Dear Dr. Hadden,

We are pleased to inform you that your manuscript 'Mechanistic insights into CAM-induced disruption of HBV capsids revealed by all-atom MD simulations' has been provisionally accepted for publication in PLOS Pathogens.

Best regards,

Michael D Robek

Academic Editor

PLOS Pathogens

Robert Kalejta

Section Editor

PLOS Pathogens

Sumita Bhaduri-McIntosh

Editor-in-Chief

PLOS Pathogens

orcid.org/0000-0003-2946-9497

Michael Malim

Editor-in-Chief

PLOS Pathogens

orcid.org/0000-0002-7699-2064

Reviewer Comments (if any, and for reference):

Reviewer's Responses to Questions

**Part I - Summary**

Reviewer #1: The authors have addressed all my concerns. Now I recommend it for publication as is.

Reviewer #2: (No Response)

Reviewer #3: The authors have addressed my comments with satisfaction!

**Part II – Major Issues: Key Experiments Required for Acceptance**

Reviewer #1: (No Response)

Reviewer #2: (No Response)

Reviewer #3: No.

**Part III – Minor Issues: Editorial and Data Presentation Modifications**

Reviewer #1: (No Response)

Reviewer #2: (No Response)

Reviewer #3: No.

PLOS authors have the option to publish the peer review history of their article (what does this mean?). If published, this will include your full peer review and any attached files.

Reviewer #1: No

Reviewer #2: No

Reviewer #3: No

---

## [Editor Report · Acceptance letter]

Dear Dr. Hadden-Parilla,

We are delighted to inform you that your manuscript, "Mechanistic insights into CAM-induced disruption of HBV capsids revealed by all-atom MD simulations," has been formally accepted for publication in PLOS Pathogens.

Best regards,

Sumita Bhaduri-McIntosh

Editor-in-Chief

PLOS Pathogens

orcid.org/0000-0003-2946-9497

Michael Malim

Editor-in-Chief

PLOS Pathogens

orcid.org/0000-0002-7699-2064